# The head direction circuit of two insect species

**Ioannis Pisokas[1]\*, Stanley Heinze[2], Barbara Webb[1]**

[1]School of Informatics, University of Edinburgh, Edinburgh, United Kingdom; [2]Lund Vision Group and NanoLund, Lund University, Lund, Sweden

**Abstract** Recent studies of the Central Complex in the brain of the fruit fly have identified neurons with activity that tracks the animal's heading direction. These neurons are part of a neuronal circuit with dynamics resembling those of a ring attractor. The homologous circuit in other insects has similar topographic structure but with significant structural and connectivity differences. We model the connectivity patterns of two insect species to investigate the effect of these differences on the dynamics of the circuit. We illustrate that the circuit found in locusts can also operate as a ring attractor but differences in the inhibition pattern enable the fruit fly circuit to respond faster to heading changes while additional recurrent connections render the locust circuit more tolerant to noise. Our findings demonstrate that subtle differences in neuronal projection patterns can have a significant effect on circuit performance and illustrate the need for a comparative approach in neuroscience.

## Introduction

For a variety of behaviours that relocate an insect in its environment, it is important for the animal to be able to keep track of its heading relative to salient external objects. This external reference object could be a nearby target, a distant landmark or even a celestial beacon. In insects, the discovery of a neuronal circuit with activity that tracks heading direction provides a potential basis for an internal compass mechanism (*Zhang, 1996*; *Homberg, 2004*; *Heinze and Reppert, 2012*). Such an internal compass can mediate a simple navigation competence such as maintaining a straight course (*Dacke et al., 2003*; *Mouritsen and Frost, 2002*) or reorienting to a target after distractions (*Neuser et al., 2008*), but is also essential for the more complex navigational process of path integration (or dead reckoning) which enables central-place foragers to return directly to their nest after long and convoluted outward paths (*Darwin, 1873*; *von Frisch, 1967*; *Mittelstaedt and Mittelstaedt, 1980*; *Müller and Wehner, 1988*). While the neural basis underlying these navigation strategies is not known in detail, a brain region called the central complex (CX) is implicated in many navigation-related processes.

The CX of the insect brain is an unpaired, midline-spanning set of neuropils that consist of the protocerebral bridge (PB), the ellipsoid body (also called lower division of the central body), the fan-shaped body (also called upper division of the central body) and the paired noduli. These neuropils and their characteristic internal organisation in vertical slices combined with horizontal layers are highly conserved across insect species. This regular neuroarchitecture is generated by sets of columnar cells, innervating individual slices, as well as large tangential neurons, innervating entire layers. The structured projection patterns of columnar cells result in the PB being organised in 16 or 18 contiguous glomeruli and the ellipsoid body (EB) in eight adjoined tiles.

Crucially, the CX is of key importance for the computations required to derive a heading signal (*Pfeiffer and Homberg, 2014*; *Triphan et al., 2010*; *Neuser et al., 2008*; *Ofstad et al., 2011*; *Homberg, 2004*; *Homberg et al., 2011*). In locusts (*Schistocerca gregaria*), intracellular recordings have revealed a neuronal layout that topographically maps the animal's orientation relative to

\*For correspondence:
i.pisokas@sms.ed.ac.uk

**Competing interests:** The authors declare that no competing interests exist.

simulated skylight cues, including polarised light and point sources of light (*Heinze and Homberg, 2007*; *el Jundi et al., 2014*; *Pegel et al., 2019*). Calcium imaging of columnar neurons connecting the EB and the PB (E-PG neurons) in the fruit fly *Drosophila melanogaster* revealed that the E-PG neuronal ensemble maintains localised spiking activity — commonly called an activity 'bump' — that moves from one group of neurons to the next as the animal rotates with respect to its surrounding (*Seelig and Jayaraman, 2015*; *Giraldo et al., 2018*). This has been confirmed for restrained flies walking on an air-supported rotating ball (*Seelig and Jayaraman, 2015*) as well as tethered flies flying in a virtual reality environment (*Kim et al., 2017*). Notably, the heading signal (the activity 'bump') is maintained even when the visual stimulus is removed, and it moves relative to the (no longer visible) cue as the animal walks in darkness (*Seelig and Jayaraman, 2015*). The underlying circuit therefore combines idiothetic and allothetic information into a coherent heading signal. Overall, this neuronal activity appears to constitute an internal encoding of heading in the insect's CX, which closely resembles the hypothetical ring attractor (*Amari, 1977*) proposed by *Skaggs et al., 1995* to account for the rat 'head direction' cells (*Taube et al., 1990*; *Blair and Sharp, 1995*; *Redish et al., 1996*; *Stackman and Taube, 1998*; *Goodridge et al., 1998*; *Goodridge and Touretzky, 2000*; *Sharp et al., 2001*; *Taube and Bassett, 2003*; *Stratton et al., 2010*). That is, the activity has the following key properties associated with ring attractors: input to the circuit results in a single localised 'bump' of activity — centred in one subset of the neurons — while other neuronal units are silenced; the activity 'bump' can move around the attractor space, which forms a ring, in a manner that consistently tracks some property of the input; and the 'bump' of activity is maintained for some time after all input is removed. These properties can be obtained, in computational neural models, by utilising opposing excitatory and inhibitory connections with excitatory lateral connections to neighbouring neuronal units and inhibitory ones affecting neurons on the opposite side of the ring.

In recent years, several computational models of the fly's CX heading tracking circuit have been presented. Some of these models are abstract while others attempt to ascribe particular roles to neurons (*Cope et al., 2017*; *Kakaria and de Bivort, 2017*; *Su et al., 2017*; *Kim et al., 2017*). *Cope et al., 2017* proposed a ring attractor model that is inspired by the rat 'head direction' cell model of *Skaggs et al., 1995*. *Kakaria and de Bivort, 2017* presented a spiking neuronal model consisting of the four types of CX neurons shown to play a role in heading encoding: E-PG, P-EN, P-EG, and Delta7 neurons. Their model demonstrated that this neuron set is sufficient for exhibiting ring attractor behaviour. In contrast, *Su et al., 2017* implemented a spiking neuronal model consisting of the E-PG, P-EN, and P-EG neurons with inhibition provided by a group of R ring neurons. In both neurobiological studies and computational models, the key neurons variously involved in the hypothetical ring attractor circuit are the E-PG, P-EN, P-EG, Delta7 and R ring neurons (*Wolff and Rubin, 2018*; *Wolff et al., 2015*; *Kakaria and de Bivort, 2017*; *Su et al., 2017*; *Green et al., 2017*; *Kim et al., 2017*). The E-PG, P-EN and P-EG neurons have been postulated to be excitatory while Delta7 or R ring neurons are conjectured to be mediating the inhibition (*Kakaria and de Bivort, 2017*; *Su et al., 2017*). The E-PG and P-EN neurons are postulated to form synapses in the PB and in the EB forming a recurrent circuit. The ring attractor state is set by a mapping of the azimuthal position of visual cues to E-PG neurons around the ring which are assumed to receive the positional input to this circuit. Furthermore, P-EN neurons shift the heading signal around the ring attractor when stimulated, in a fashion similar to the left-right rotation neurons proposed by *Skaggs et al., 1995* (*Turner-Evans et al., 2017*; *Green et al., 2017*). In principle, two main types of ring attractor implementation exist: one with local excitation and global, uniform, inhibition and another one characterised by sinusoidally modulated inhibition across the ring attractor. *Kim et al., 2017* have experimentally explored the type of ring attractor that could underlie the head direction circuit of the fruit fly and concluded that the observed dynamics of E-PG neurons can best be modelled using a ring attractor with local excitation and uniform global inhibition.

The above-outlined overall circuit depends critically on the detailed anatomical connections between cell types of the CX, so that the implementation of a specific type of ring attractor imposes additional constraints on the neuronal connection patterns and individual morphologies. Although the CX is highly conserved on a broad level, details at the level of single neurons vary between insect species. Yet, conclusions about the function of the circuit are usually drawn from *Drosophila* data and applied to insects in general. Given numerous differences in the CX neuroarchitecture between insects, we asked whether a ring attractor circuit is also plausible when taking into account anatomical data from another model species, the desert locust.

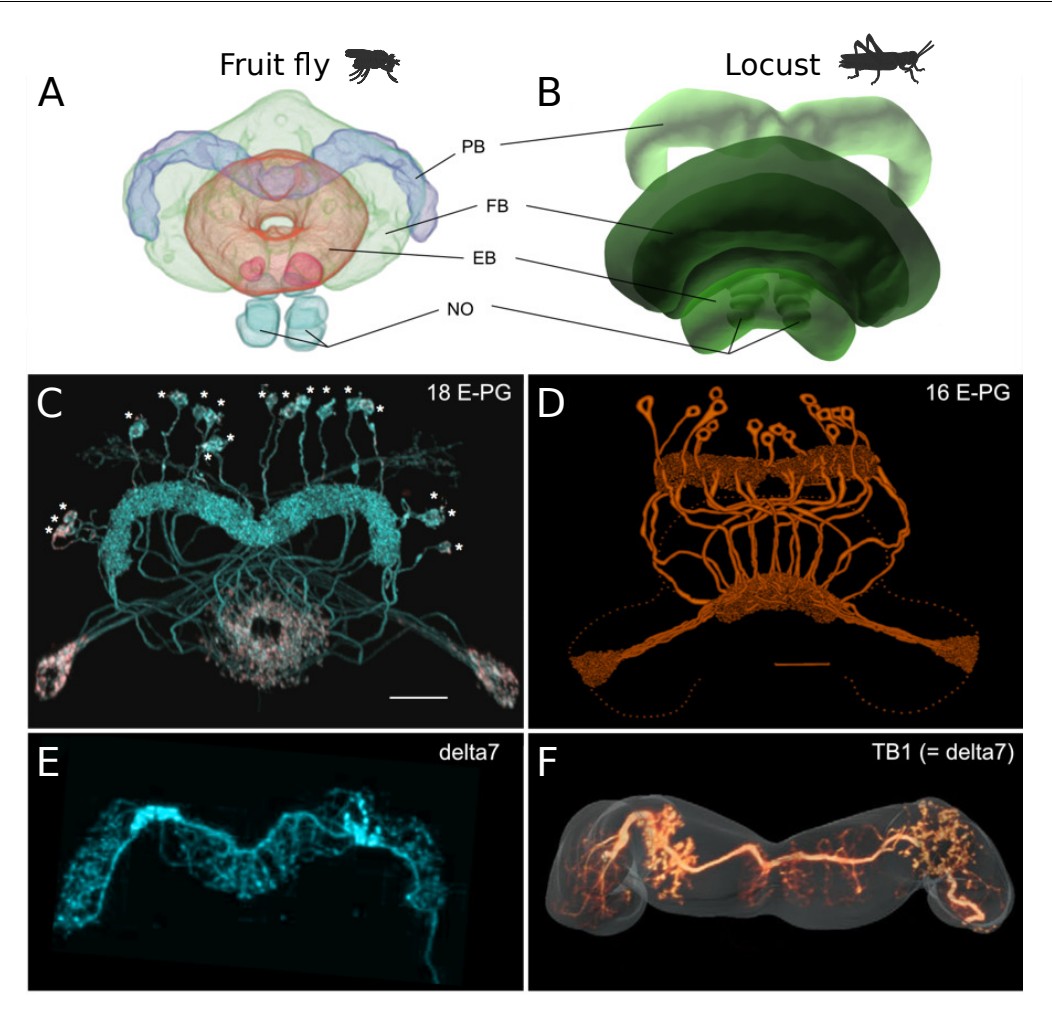

**Figure 1.** Anatomical differences between two species. There are three apparent differences between the CX of the fruit fly (*Drosophila melanogaster*) and the desert locust (*Schistocerca gregaria*). (**A**, **B**) The ellipsoid body in the fruit fly has a toroidal shape while in the locust is crescent-shaped so its two ends are separate. (**C**, **D**) The protocerebral bridge consists of 18 glomeruli and 18 corresponding E-PG and P-EG neurons in the fruit fly (see *Table 3*) while in the locust there are 16 glomeruli and neurons innervating them. (**E**, **F**) The Delta7 neurons in the fruit fly have postsynaptic domains along the whole length of their neurite while in the desert locust only in specific sections with gaps in between.

© 2018 Wiley Periodicals, Inc. Panel A, C and E are reproduced and adapted from *Wolff and Rubin, 2018* with permission from Wiley Periodicals, Inc. They are not covered by the CC-BY 4.0 licence and further reproduction of this panel would need permission from the copyright holder.

© 2009 Insect Brain Database. Panel B is an original image only available for non-commercial use from *El Jundi et al., 2009* with permission from Insect Brain Database, https://insectbraindb.org. They are not covered by the CC-BY 4.0 licence and further reproduction of this panel would need permission from the copyright holder.

© 1998 Wiley-Liss, Inc. Panel D is reproduced from *Vitzthum and Homberg, 1998* with permission from Wiley-Liss, Inc. They are not covered by the CC-BY 4.0 licence and further reproduction of this panel would need permission from the copyright holder.

© 2015 Wiley Periodicals, Inc. Panel F is reproduced from Figure 1J of *Beetz et al., 2015* with permission from Wiley Periodicals, Inc. They are not covered by the CC-BY 4.0 licence and further reproduction of this panel would need permission from the copyright holder.

The online version of this article includes the following figure supplement(s) for figure 1:

**Figure supplement 1.** Connectivity matrices of the two species.

Three main differences are evident when comparing the CX of the fruit fly and the locust (*Figure 1*). First, as in most insects except *Drosophila*, the EB of the locust is not closed around the edges, but is crescent-shaped, preventing the E-PG neurons from forming a physical ring. Second, the *Drosophila* PB consists of nine glomeruli per hemisphere, and accordingly 18 groups of E-PG neurons. In locusts, there are 8 glomeruli per hemisphere and 16 groups of neurons. Third, a key part of the proposed ring attractor circuit, the Delta7 neurons (TB1 neurons in the locust) differ strikingly in their arborization pattern across the width of the PB. Whereas these cells possess two columnar output sites located eight glomeruli apart in all species, their dendrites have an approximately uniform density across the PB glomeruli in *Drosophila*. This differs substantially from the dendritic distribution in the desert locust, in which the postsynaptic domains of the eight Delta7 neurons are restricted to particular glomeruli of the PB, avoiding the regions around the output branches. This pattern is conserved in other species as well, such as in the Monarch butterfly (*Danaus plexippus*), the sweat bee (*Megalopta genalis*), as well as in two species of dung beetles (*Scarabaeus lamarcki* and *Scarabaeus satyrus*) (*Heinze and Homberg, 2007*; *Heinze et al., 2013*; *Stone et al., 2017*; *El Jundi et al., 2018*). Given these three differences of the *Drosophila* CX from other insects, we explored the functional consequences of each difference and how these might relate to the behavioural characteristics of each insect.

To explore this question, we used the anatomical projection patterns of the main CX neuron types in flies and locusts and derived the effective neuronal circuits by simplifying anatomical redundancy. Both resulting circuits indeed have the structural topology of a ring attractor. Despite significant anatomical differences the homologous circuits in the fruit fly and the locust are structurally similar but not identical. Their differences have significant functional effect in the ability of the two circuits to track fast rotational movements and to maintain a stable heading signal. Our results highlight that even seemingly small differences in the distribution of dendritic fibres can affect the behavioural repertoire of an animal. These differences, emerging from morphologically distinct single neurons, highlight the importance of a comparative approach to neuroscience. Rather than assuming results from model species are generalisable, we gain deeper insight into function by discovering which elements are actually shared across species and what are the consequences of observed variation.

## Results

### The effective circuit

The neuronal projection data of the fruit fly and the desert locust were encoded in connectivity matrices and used for the simulations we report here (*Wolff et al., 2015*; *Wolff and Rubin, 2018*; *Heinze and Homberg, 2007*; *Heinze and Homberg, 2008*; *Heinze and Homberg, 2009*; *Heinze et al., 2009*). While some simplifications could not be avoided, we have exclusively used projection patterns grounded in anatomical data for each species to construct the connectivity matrices. To facilitate conceptual understanding, we visualised the connectivity matrices as directed graphs and analysed the effective connectivity of the neuronal components of the CX for both species.

#### Inhibitory circuit

First, we focus on the inhibitory portion of the circuit. Study of the actual neuronal anatomy of Delta7 neurons in the PB shows that, in both species, each Delta7 neuron has presynaptic terminal domains in two or three glomeruli along the PB (*Heinze and Homberg, 2007*; *Wolff and Rubin, 2018*). These presynaptic terminal domains are separated by seven glomeruli (*Figure 2A* and *Figure 2D*). In *Drosophila*, the Delta7 neurons have postsynaptic terminals across all remaining glomeruli of the PB (*Wolff and Rubin, 2018*; *Franconville et al., 2018*) while in locusts the Delta7 neurons have postsynaptic terminal domains only in specific glomeruli (*Heinze and Homberg, 2007*; *Beetz et al., 2015*; *Hadeln et al., 2020*).

There are eight types of Delta7 neurons in the PB, each having the same pattern of synaptic terminals shifted by one glomerulus (*Figure 2A* and *Figure 2D*). Within each glomerulus, the Delta7 neuron with presynaptic terminals is assumed to form synapses with all other Delta7 neurons that have postsynaptic terminals in the same glomerulus. Since each Delta7 neuron is presynaptic to the same Delta7 neurons in two or three glomeruli along the PB, we reduce these two or three synaptic

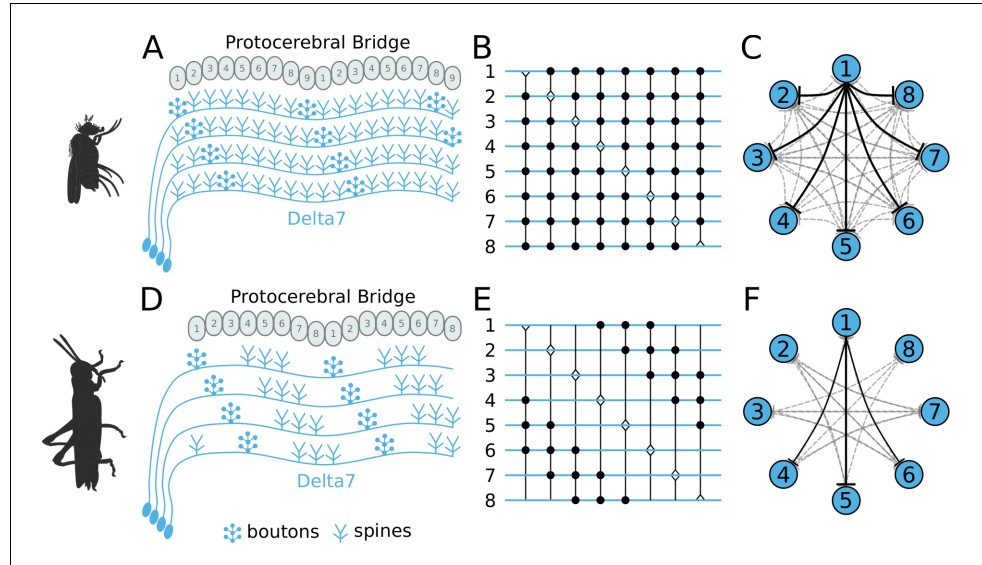

**Figure 2.** Effective connectivity of the inhibitory (Delta7) neurons. On the top row is the fruit fly circuit, on the bottom row is the locust circuit. In A and D, four examples of how the eight types of Delta7 neurons innervate the PB are illustrated. In both species, presynaptic domains are separated by seven glomeruli. (**B** and **E**) Effective connectivity. Each horizontal blue line represents one Delta7 neuron. Vertical lines represent axons, with triangles indicating outputs from Delta7 neurons and filled circles representing inhibitory synapses between axons and other Delta7 neurons. (**C** and **F**) Alternative depiction of the circuit in graph form with blue circles representing Delta7 neurons and lines representing inhibitory synapses between pairs of neurons. Each Delta7 neuron inhibits all other Delta7s in the fruit fly (**C**), but only more distant Delta7s in the locust (**F**).

domains to one single equivalent synapse between each pair of Delta7 neurons in order to draw a simplified equivalent circuit (*Figure 2B* and *Figure 2E*). In order to highlight the main functional differences, we redrew these neuronal circuits in a network graph form which revealed an eight-fold radial symmetry in both species, regardless of the different neuronal anatomies and the anatomical presence of nine PB glomeruli in flies.

The network graph form of the circuit further makes evident a global, uniform, inhibition pattern in the case of the fruit fly versus a local inhibition pattern in the case of the locust (*Figure 2C* and *Figure 2F*). That is, in fruit flies each Delta7 neuron forms synapses and inhibits all other Delta7 neurons. In contrast, in the locust each Delta7 neuron only inhibits a subset of Delta7 neurons with weakening synaptic strengths towards its nearest neighbours (*Heinze and Homberg, 2007*). The effective global inhibition pattern found in the fruit fly fits the observation of *Kim et al., 2017* that calcium dynamics better matched a ring attractor with global inhibition in this species.

## Excitatory circuit

We next focused on the excitatory portion of the hypothetical ring attractor circuit. For deriving the effective circuit of the excitatory portion of the network, it was necessary to employ an unconventional numbering scheme for the PB glomeruli; that is, in both hemispheres, glomeruli are numbered incrementally from left to right, 1–9 for the fruit fly (*Figure 3*) and 1–8 for the locust (*Figure 4*). EB tiles were numbered 1 to 8 for both species. For brevity, throughout this text, we denote a tile numbered '1' as T1 and a glomerulus numbered '1' as G1. Neurons are numbered by the glomerulus they innervate, using a numerical subscript, e.g. P-EN$_1$ for the P-EN neurons innervating glomeruli G1.

In accordance with calcium imaging (*Turner-Evans et al., 2017*; *Green et al., 2017*), simulating the fruit fly and locust circuits confirmed that there are two activity 'bumps' along the PB. The choice of unconventional numbering scheme for the PB glomeruli has as an effect that both activity 'bumps' are centred around neurons innervating identically numbered glomeruli (*Figure 3—figure supplement 1*). We use this symmetry to simplify the circuit and derive the effective connectivity.

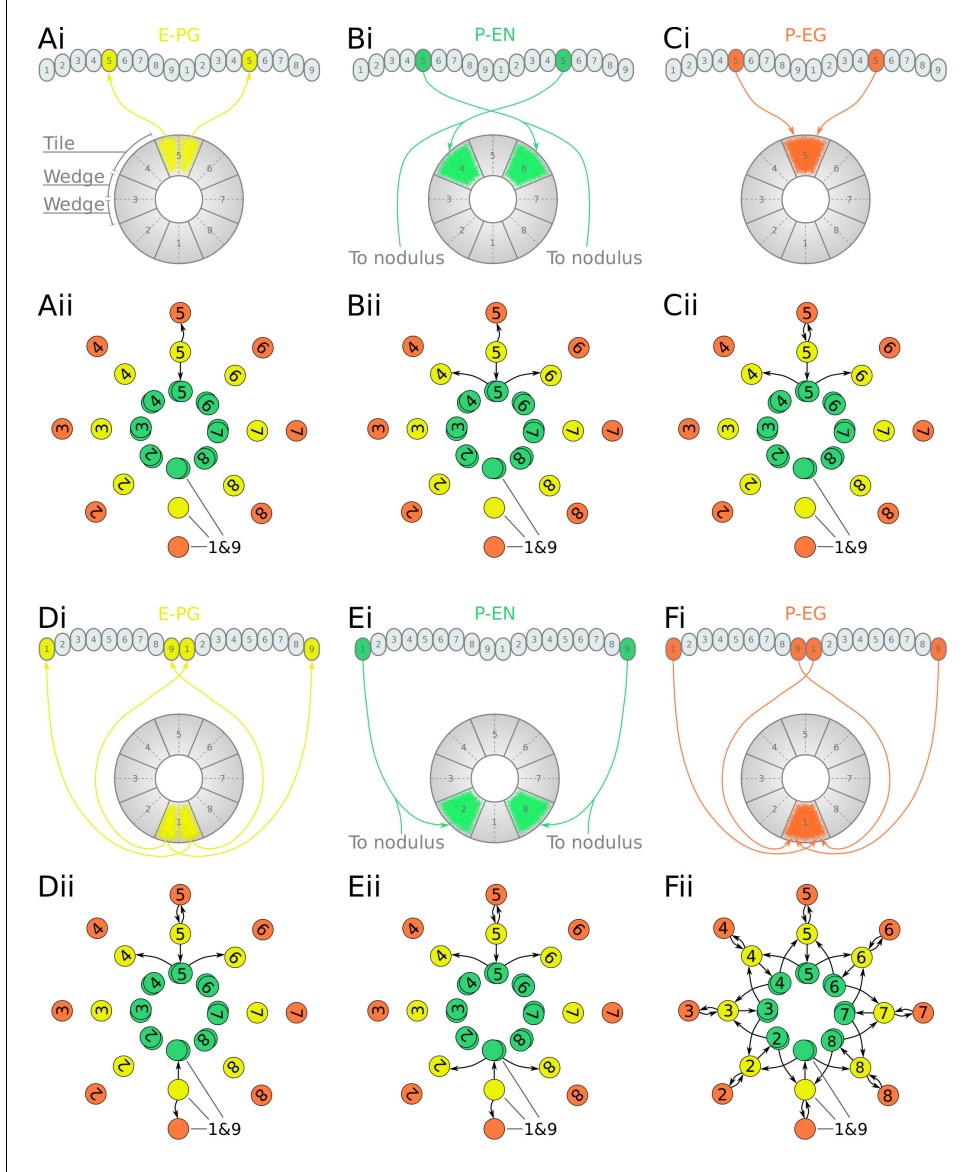

**Figure 3.** Projection patterns of the excitatory portion of the fruit fly circuit. (**Ai–Fi**) Examples of E-PG (combined E-PG and E-PG$_T$, see **Table 3**), P-EN and P-EG neurons with their synaptic domains and projection patterns. (**Aii–Cii**) Step by step derivation of the effective circuit as a directed graph network (see main text for a complete description). Each coloured disc represents a group of neurons with arrows representing excitatory synaptic connections. Pairs of E-PG and P-EG neurons can be considered to act as single units connecting the respective tile to equally numbered PB glomeruli in both hemispheres, while P-EN neurons are shown overlapped because each receives input only from its contralateral nodulus. (**Dii–Eii**) The connectivity also allows neurons innervating glomeruli 1 and 9 to act as a single unit. (**Fii**) Depiction of the complete effective connectivity of the excitatory circuit, which has an eight-fold radial symmetry.

The online version of this article includes the following video and figure supplement(s) for figure 3:

**Figure supplement 1.** Neuronal activity across PB glomeruli.

**Figure supplement 2.** Neuronal projections in the fruit fly.

**Figure 3—video 1.** Animation illustrating the operation of the excitatory portion of the fruit fly circuit.

https://elifesciences.org/articles/53985#fig3video1

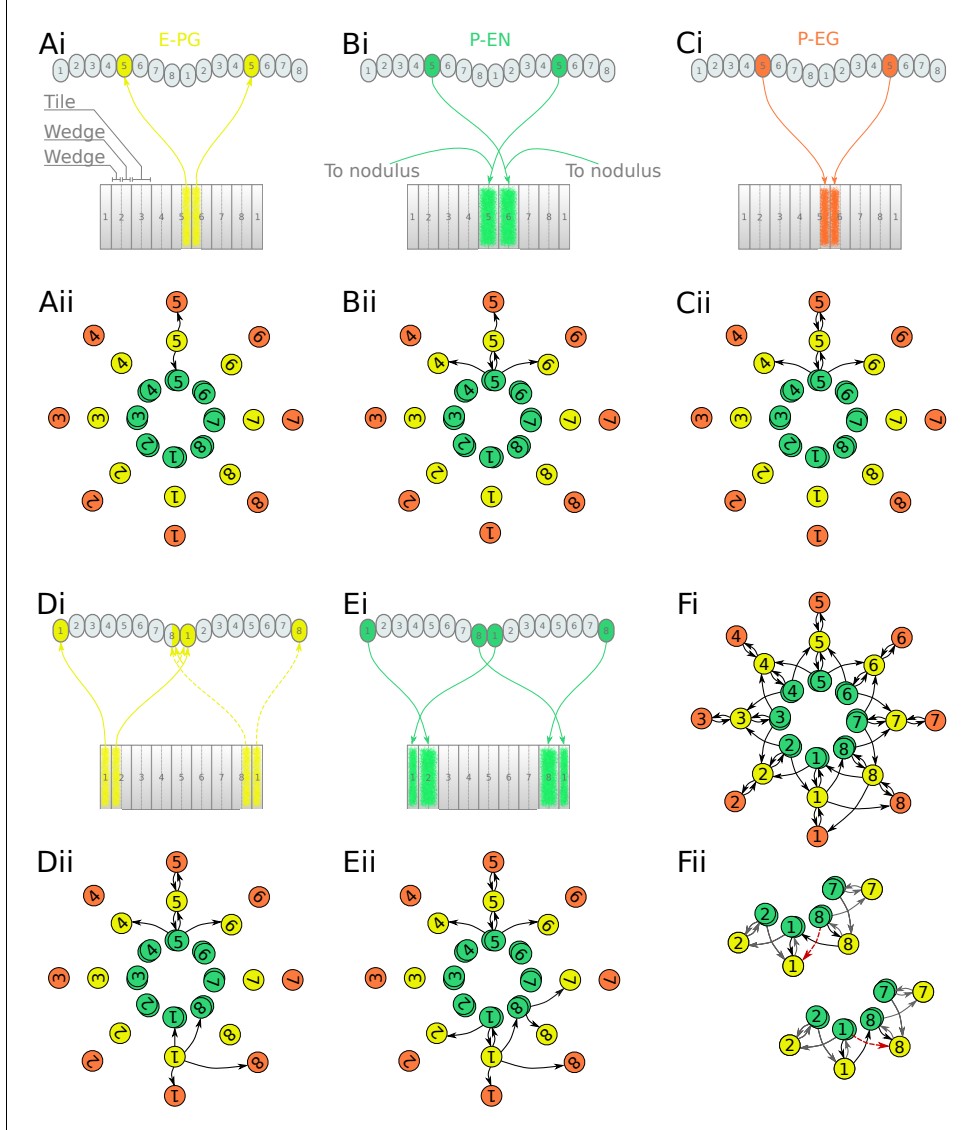

**Figure 4.** Projection patterns of the excitatory portion of the locust circuit. (**Ai–Ei**) Examples of E-PG, P-EN and P-EG neurons with their synaptic domains and projection patterns. (**Aii–Eii**) Step by step derivation of the effective circuit (see main text for a complete description). Each coloured disc represents a group of neurons with arrows representing excitatory synaptic connections. Pairs of E-PG and P-EG neurons can be considered to act as single units connecting the respective tile to equally numbered PB glomeruli in both hemispheres, while P-EN neurons are shown overlapped because each receives input only from its contralateral nodulus. Note that the numbering of the EB slices is conceptual and arbitrary, chosen to assist description of the circuit organisation; what matters for the connectivity is the overlap of the synaptic domains in the EB and not the particular numbering choice. (**F**i) The complete effective connectivity of the locust excitatory circuit closely resembles that of the fruit fly. (**Fii**) Between octants 1 and 8, the locust circuit obtains functional connectivity from P-EN$_8$ to 'neighbouring' E-PG$_1$ (red dashed arrow) via three actual connections: P-EN$_8$ to E-PG$_8$ to P-EN$_1$ to E-PG$_1$ (black arrows); and equivalently for P-EN$_1$ to E-PG$_8$.

The online version of this article includes the following video and figure supplement(s) for figure 4:

**Figure supplement 1.** Neuronal activity across PB glomeruli.

**Figure supplement 2.** Neuronal projections in the locust.

**Figure 4—video 1.** Animation illustrating the operation of the excitatory portion of the locust circuit.

https://elifesciences.org/articles/53985#fig4video1

First, we analyse and derive the effective circuit of the fruit fly. Under our numbering scheme, each E-PG neuron has synaptic domains in identically numbered EB tiles and PB glomeruli (e.g. *Figure 3Ai*). That is, E-PG$_5$ neurons have synaptic domains in tile T5 and glomeruli G5 in both hemispheres of the PB. Since activity is symmetrical in both PB hemispheres, the pair of E-PG$_5$ neurons forms a single functional unit, as illustrated in the equivalent circuit (*Figure 3Aii*), with single synaptic connections shown to the corresponding P-EN$_5$ and P-EG$_5$ neurons. P-EN neurons, however, connect corresponding glomeruli from each PB hemisphere to two tiles, one shifted to the left and one to the right, for example, P-EN$_5$ would connect glomeruli G5 to tiles T4 and T6 (*Figure 3Bi*). P-EN neurons are indicated as two overlapped discs in the equivalent circuit (*Figure 3Bii*) because even though each pair receives common input in the glomeruli they also receive differential angular velocity input, depending on the hemisphere they innervate. The pair of P-EN$_5$ neurons forms synapses with E-PG$_4$ neurons in T4 and E-PG$_6$ neurons in T6, respectively. A third class of cells, P-EG neurons, innervate equally numbered glomeruli and tiles, following the same pattern as the E-PG neurons but with their presynaptic and postsynaptic terminals on opposite ends (*Figure 3Ci*), and are thus illustrated as single functional units in the equivalent circuit (*Figure 3Cii*). Following the synaptic connections forward around the circuit (*Figure 3—figure supplement 2*), E-PG$_4$ and E-PG$_6$ neurons innervate glomeruli G4 and G6 respectively, forming synapses with P-EN and P-EG neurons in these glomeruli; P-EG$_6$ make reciprocal connections to E-PG$_6$; and the paired P-EN$_6$ neurons make connections back to T5 and onward to T7, etc. Thus the connectivity pattern shown in *Figure 3Cii* is repeated all the way around the circuit. Crucially, tile T1 is innervated by both E-PG$_1$ and E-PG$_9$ which also innervate glomeruli G1 and G9, respectively (*Figure 3Di*). These neurons can also be treated as one unit, E-PG$_{1\&9}$, in the effective circuit (*Figure 3Dii*) because they receive common synaptic input. Since there are no P-EN neurons innervating the innermost glomeruli (G9 in the left and G1 in the right hemisphere), P-EN$_1$ and P-EN$_9$ consist a pair of neurons in the equivalent circuit, making onward connections to tiles T2 and T8, and thus E-PG$_2$ and E-PG$_8$, respectively (*Figure 3Ei and Eii*). Therefore, the effective circuit of the fruit fly has an eight-fold radial symmetry despite the nine PB glomeruli (illustrated in *Figure 3Fii*).

We follow a similar procedure to derive the effective circuit in the locust (*Figure 4*). Here, E-PG neurons from the two corresponding PB glomeruli, one in each hemisphere, have synaptic domains in two neighbouring EB wedges (half tiles), for example E-PG$_5$ innervates two wedges in tiles T5 and T6 and glomeruli G5 of the PB (*Figure 4Ai* and *Figure 4—figure supplement 2*). Note that in the equivalent circuit (*Figure 4Aii*) we label these neurons by the relevant glomeruli number '5' and can still treat them as a single unit connecting (as for the fruit fly) to the P-EN$_5$ and P-EG$_5$. P-EN neurons connect PB glomeruli to tiles shifted by one wedge to the left and right, for example glomeruli G5 with tiles T5 and T6 (*Figure 4Bi*). This is a shift of half tile while in the fruit fly we see a whole tile shift. As a consequence, P-EN$_5$ neurons effectively make reciprocal connections back to E-PG$_5$, which does not occur in the fruit fly. However, similar to the fruit fly, the P-EN$_5$ neurons also make onward connections to E-PG$_6$ and E-PG$_4$ (note that following the same labelling system as above, the E-PG$_4$ innervates neighbouring wedges in T4 and T5) (*Figure 4Bii*). Finally, P-EG neurons follow a similar pattern to E-PG neurons (*Figure 4Ci*), innervating equally numbered glomeruli and two wedges in neighbouring tiles, e.g. P-EG$_5$ connects G5 to T5 and T6, which can be shown as a single unit making a reciprocal connection to E-PG neurons with the same number. Tracing this connectivity pattern forward as before, the connections are repeated around the circuit. The circuit forms a closed ring because the pair of E-PG neurons innervating the medial glomeruli (glomerulus G8 in the left and G1 in the right hemisphere) have arborizations that cross the borders of these two glomeruli (*Heinze and Homberg, 2009*, *Figure 1*) hence forming synapses with both P-EN$_1$ and P-EN$_8$ neurons in the two medial glomeruli (*Figure 4Di,Ei*). This evolutionary adaptation results in a closed ring without the need for an extra pair of neurons connecting the two edges of the EB. The crossing of glomeruli borders is characteristically evident in these two medial glomeruli resulting in a modified connectivity pattern between octants 1 and 8 of the circuit (*Figure 4Fi*). Even though this pattern might at first appear to break the structural radial symmetry in effect it provides a functional continuity of left-right activity 'bump' shifting all around the ring as illustrated in *Figure 4Fii*. *Figure 4Fii* shows in detail the specific portion of *Figure 4Fi*, illustrating how the connectivity we found in the animal effectively functions equivalently to the other P-EN to E-PG connections around the ring. The dashed red arrows show the effective connections closing the ring.

In spite of the EB in the locust not being torus-shaped but rather having a crescent shape, the effective circuit still forms a closed ring with an eight-fold structure almost identical to that of the fruit fly (*Figure 4Fi*). This is a consequence of the combination of E-PG neurons selectively cross-innervating the two medial glomeruli and the P-EN neurons forming reciprocal connections back to the E-PG neurons in the same octant. Both of these features are missing in the fruit fly. We thus observe the existence of two different solutions to the same problem, in the fruit fly the torus-shaped EB anatomically facilitates closing the ring while in the locust, which has an EB with open ends, adaptations in the neuronal projection patterns result again in a closed ring.

## Overall circuit

The similarity between the effective circuits of the locust and the fruit fly is striking. Despite the fact that locusts have eight PB glomeruli while fruit flies have nine, both circuits form closed rings organised in eight octants with the functional role of each neuron class appearing to be identical. The E-PG neurons were presynaptic to both P-EG and P-EN neurons, with P-EG neurons forming recurrent synapses back to E-PG neurons. P-EN neurons were presynaptic to E-PG neurons with a shift of one octant to the left or right. Overall, two of the main anatomical differences between the two species (eight versus nine PB glomeruli and ring-shaped versus crescent-shaped EB) had no fundamental effect on the principal structure of the CX heading direction circuit.

During our analysis of the anatomical data in locusts and flies, we observed that the order of E-PG neuronal projections in the EB differs between the two species (*Heinze and Homberg, 2008*; *Williams, 1975*; *Wolff et al., 2015*; *Wolff and Rubin, 2018*). Spanning the EB clockwise starting from tile 1, the fruit fly wedges connect first to the right PB hemisphere, then to the left and so on, while in the locust they connect first to the left, then to the right and so on. However, despite this seemingly major difference in projection patterns the effective circuit is preserved between the two species.

The excitatory portions of the circuits differed in that the locust P-EN neurons make synapses back to E-PG neurons in the same octant while in the fruit fly they do not (Compare *Figure 3F* with *Figure 4F*). This difference resulted from the P-EN synaptic domains being shifted by half-tile in the locust instead of the whole tile shift seen in the fruit fly (*Figure 3B* and *Figure 4B*). Consequently, the middle portion of neighbouring P-EN synaptic domains overlap in the EB and feed back to E-PG neurons in the same octant of the ring. This specialisation of the locust together with the cross-innervation of the two medial glomeruli by E-PG neurons enable the closing of the ring in the locust.

When we combined the inhibitory and the excitatory sub-circuits into a complete model (*Figure 5*), the E-PG neurons became presynaptically connected to the Delta7 neurons, in line with (*Franconville et al., 2018*; *Turner-Evans et al., 2019*). Additionally, each Delta7 neuron inhibits the P-EN and P-EG neurons in the same octant, as well as all other Delta7 neurons (for the fruit fly) or a subset (for the locust), as described above. This difference results in two different types of ring attractor topology; one with global inhibition in the fruit fly and another with local inhibition in the locust.

## Predicted synaptic strengths

We next focused on whether and how the two circuits could operate as ring attractors. To this end, we implemented computational models of the two circuits using neuronal projection patterns derived from the anatomical data and investigated what synaptic connectivity strengths would be required for the circuits to produce ring attractor dynamics. The results constitute a prediction for the synaptic efficacies we expect to be observed in insects when such measurements become available.

We used spiking Leaky Integrate and Fire neuron models following the same approach as *Kakaria and de Bivort, 2017* and we ran an optimisation algorithm to find regularities in the synaptic efficacy patterns that resulted in functional ring attractors (see section Materials and methods). A functional ring attractor should maintain a 'bump' of activity along the neurons of the ring, with characteristics defined in section Materials and methods. A k-means algorithm was used to identify the clusters around which solutions were found. These clusters were ordered by the number of instances found by repetitive runs of the optimiser. Although the absolute synaptic strengths are arbitrary, as they depend on unknown biophysical properties of the involved neurons, a pattern emerged in the

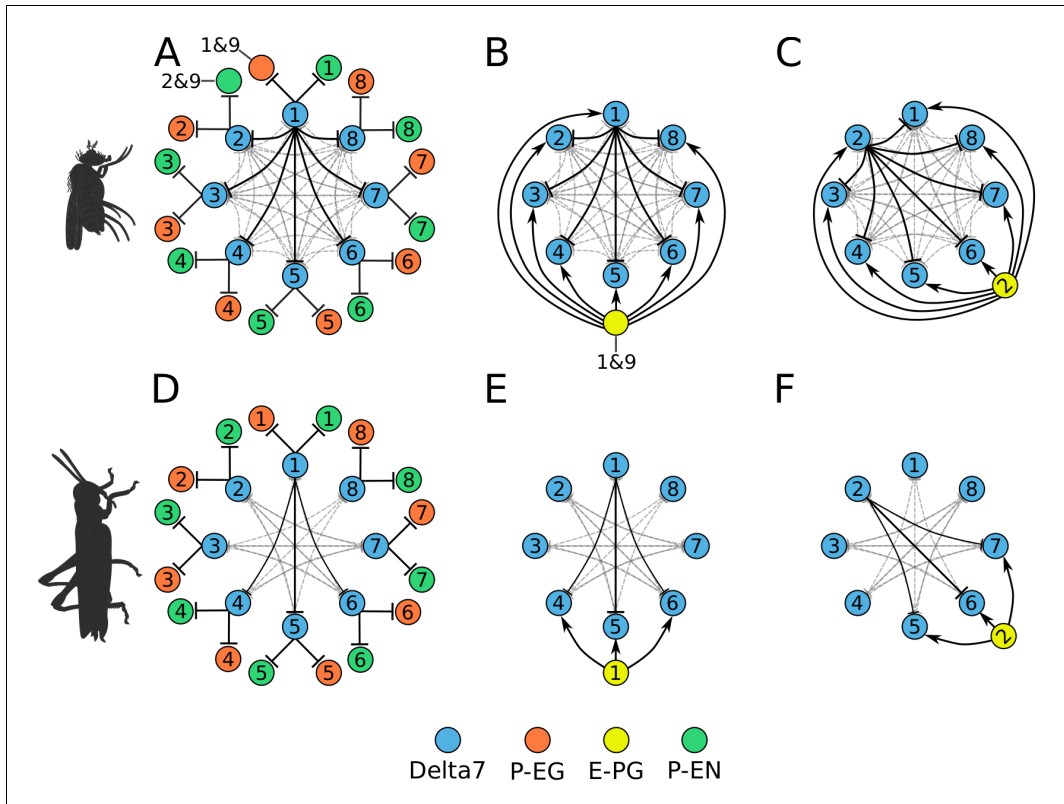

**Figure 5.** Combined excitatory and inhibitory portion of the ring attractors. Explanatory drawings of the connectivity of the inhibitory portion with the excitatory portion of the circuit for the fruit fly (**A–C**) and the locust (**D–F**). Each coloured disc represents one or more neurons with lines representing synaptic connections. (**A**) Conceptual depiction of effective global inhibition in the fruit fly. The connectivity of E-PG neurons is shown for two neurons only (**B,C** and **E,F**). In this conceptual effective connectivity drawing, E-PG neurons appear to be located on the one side of the ring making synapses around the ring. However, anatomically each E-PG neuron innervates one glomerulus where it makes all its synapses with postsynaptic Delta7 neurons that run along the PB.

relative synaptic strengths between the different synapses (*Figure 6*). The most frequent synaptic strengths patterns were comparably consistent for the fruit fly and the locust. In both species, among the excitatory synapses, the P-EN to E-PG and P-EG to E-PG synaptic strengths were the weakest, while the synaptic strengths from E-PG to P-EG and P-EN neurons were the strongest. The inhibitory synaptic strengths from Delta7 to P-EN and P-EG were stronger in the locust than in the fruit fly, which was consistent with the fly neurons receiving input from more Delta7 neurons.

## Predicted neuronal activity

Whereas our simulations confirmed that both the fruit fly and the locust circuit can operate as ring attractors, there were clear differences in the spiking activity and dynamics of the two circuits (*Figure 7*). One major difference was that Delta7 neurons exhibited distinct firing patterns in the two species. In the locust, there was a strong heading-dependent modulation in the firing of Delta7 neurons, in line with the heading signal (activity 'bump') location. Those Delta7 neurons corresponding to the current heading signal location remained silent. In contrast, in the fruit fly the firing of action potentials was only minimally modulated across the Delta7 population (*Figure 7A* and *Table 1*). This difference reflected the utilisation of local inhibition in the case of the locust versus the global inhibition in the fruit fly. Electrophysiologists have indeed reported this pronounced firing rate variation in the locust (*Heinze and Homberg, 2007*; *Heinze et al., 2009*; *Bockhorst and Homberg, 2015*; *Pegel et al., 2018*). It will be interesting to see if the fruit fly neurons indeed show a lower modulation as predicted by our model.

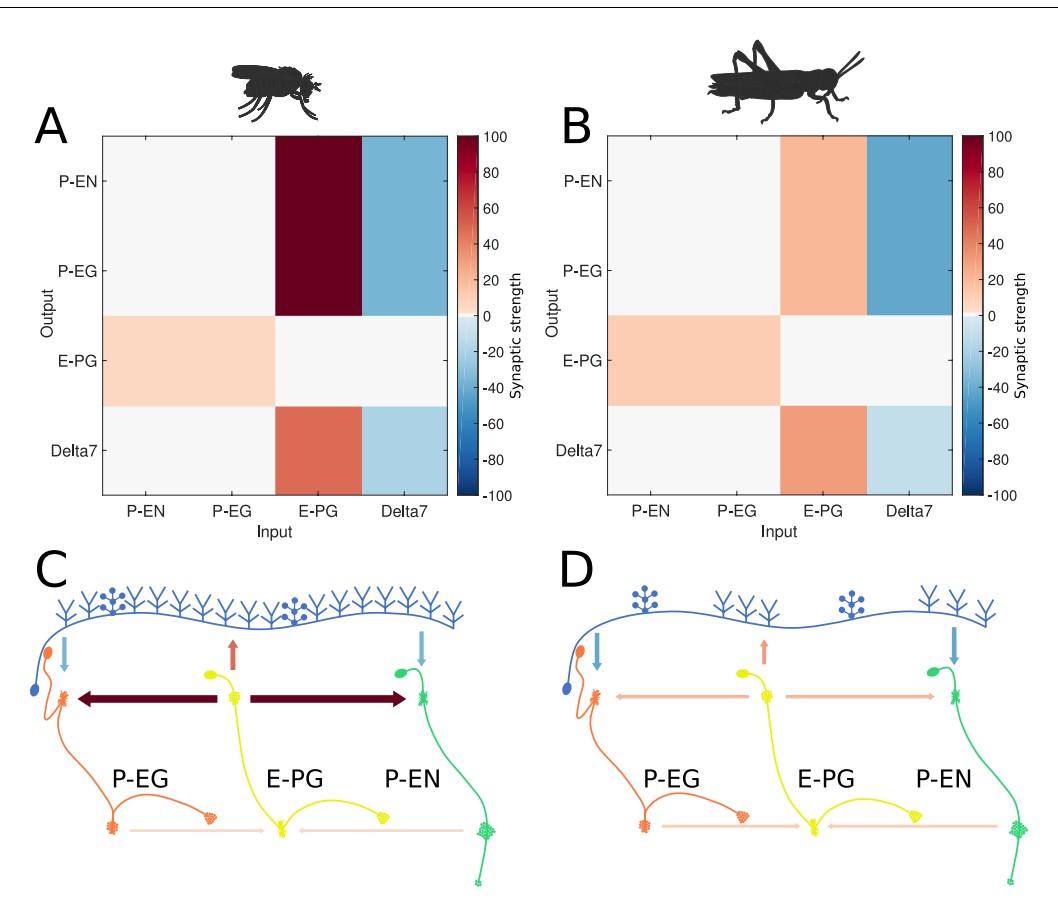

**Figure 6.** Relative synaptic strengths. Graphical depiction of the synaptic strengths between classes of neurons. (**A,C**) For the fruit fly ring attractor circuit. (**B,D**) For the desert locust ring attractor circuit. Synaptic strengths are denoted by colour in panels A and B. In panels C and D, synaptic strengths between neurons are indicated by arrow colour and thickness in scale. Note that in the locust the synaptic strengths shown for Delta7 neurons are the peak values of the Gaussian distributed strengths shown in *Figure 1—figure supplement 1*.

When comparing the head-direction tuning widths between the two species, we noted that in locusts all cell types are consistently tuned more narrowly (ca. 20%, *Table 2*). Within both species, the activity 'bump' is wider for E-PG neurons than for the other excitatory neuron classes (*Table 1*), a difference that is more pronounced in the fruit fly. The tuning of the Delta7 neurons is the widest across cell types in both species (approx. 96˚ in the locust, *Table 2*). In the fruit fly, the activity is approximately even across all Delta7 neurons (ca. 10% modulation).

In our models, we employed one neuron for each connection, whereas in the actual animals there are multiple copies of each neuron. While definite numbers of neurons will have to await electron microscopy data, there are likely at least two copies of E-PG, P-EG and P-EN neurons in each columnar module, and three to four copies of Delta7 cells (*Williams, 1975*; *Heinze and Homberg, 2008*; *Beetz et al., 2015*; *Wolff et al., 2015*; *Wolff and Rubin, 2018*). If we were to replace each modelled neuron by a bundle of neurons, the action potential firing rates shown in *Table 1* would be divided among the neurons in each bundle. The peak firing rate of each neuron would be in the range of 40–90 impulses/s which is similar to the range of the rates recorded electrophysiologically in the locust (*Heinze and Homberg, 2009*). The objective function did not explicitly constrain the firing rates of the neurons but the synergy of biophysical parameters, circuit structure and performance requirements produced working circuits that operate in firing rates similar to those recorded electrophysiologically (see section Discussion).

The steady state peak spiking rate for each group of neurons differs between the fruit fly and the locust circuits. On average, the locust neurons showed ca. 25% higher peak firing rates compared to

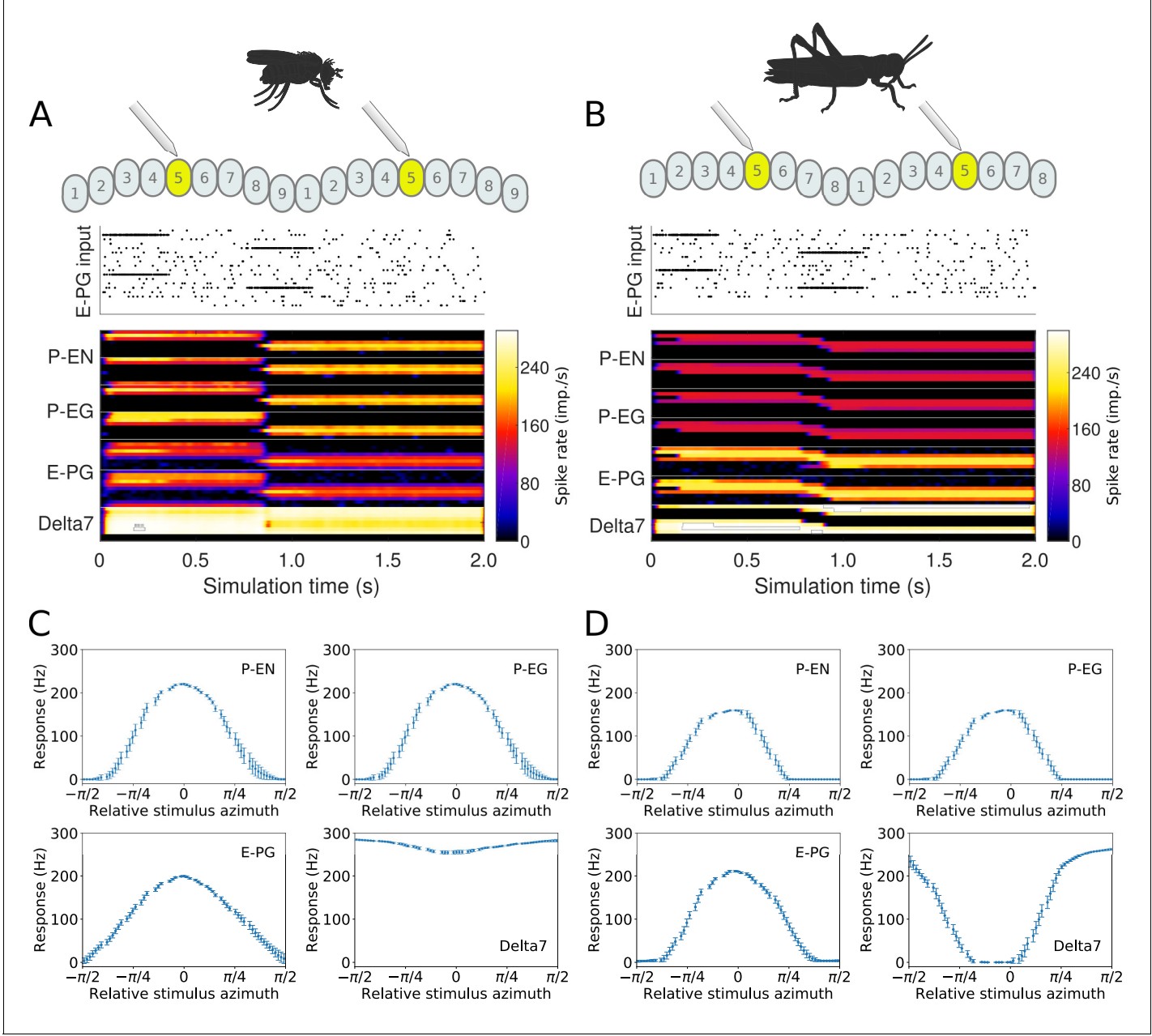

**Figure 7.** Response to abrupt stimulus changes and tuning curves of neurons. (**A** and **B**) The raster plots of the stimuli used to drive the ring attractor during the simulation are shown on top and the spiking rate activity of each neuron at the bottom. In the beginning of the simulation the stimulus spiking activity sets the ring attractor to an initial attractor state. A 'darkness' period of no stimulus follows. Then a second stimulus corresponding to a sudden change of heading by 120° is provided. In the lower parts of A and B, the spiking activity of each neuron, filtered along the time axis by a Gaussian low-pass filter with window of 120*ms* and $\sigma = 24ms$, is shown colour coded. The order of recorded neurons is the same as shown in the connectivity matrices (*Figure 1—figure supplement 1*). (**A**) Response of the fruit fly ring attractor to sudden change of heading. (**B**) Response of the locust ring attractor to sudden change of heading. Even though the activity 'bump' in the locust model tends to start transitioning sooner, the fruit fly model completes the transition faster. (**C** and **D**) Response of individual neuron types to different stimuli azimuths (n = 40 trials in each condition). The mean and standard deviation are indicated by the error bars at the sampled azimuth points. Peak activity has been shifted to 0°. (**C**) Tuning curves for the fruit fly and (**D**) tuning curves for the locust.

The online version of this article includes the following figure supplement(s) for figure 7:

**Figure supplement 1.** Response of spiking and rate-based models to step change of heading.

**Table 1.** Characteristics of the activity 'bump'.

The Full Width at Half Maximum (FWHM), the peak impulse rate of the activity 'bump' formed across each family of neurons and the amplitude of the activity 'bump' measured as the range of firing rates are shown. Measurements were made 10 s after the stimulus was removed. Numbers are given as median and standard deviation. The activity of Delta7 neurons in *Drosophila* is approximately even, hence the corresponding FWHM measurement is not meaningful and marked as 'N/A'.

| Neuron class | *Drosophila* | | | Locust | | |
|---|---|---|---|---|---|---|
| | FWHM | Peak | Amplitude | FWHM | Peak | Amplitude |
| | (°) | (imp./s) | (imp./s) | (°) | (imp./s) | (imp./s) |
| E-PG | 88.3 ± 0.3 | 161.0 ± 0.2 | 160.1 ± 0.3 | 68.3 ± 0.1 | 192.6 ± 0.1 | 192.0 ± 0.2 |
| P-EN | 80.4 ± 0.4 | 190.1 ± 0.2 | 190.1 ± 0.2 | 63.1 ± 0.3 | 153.5 ± 0.1 | 153.5 ± 0.1 |
| P-EG | 71.0 ± 0.2 | 190.1 ± 0.2 | 190.1 ± 0.2 | 63.1 ± 0.3 | 153.5 ± 0.1 | 153.5 ± 0.1 |
| Delta7 | N/A | 274.7 ± 0.1 | 27.1 ± 0.2 | 101.1 ± 0.2 | 266.6 ± 0.2 | 266.6 ± 0.2 |

the fruit fly neurons while the Delta7 neurons have the highest spiking rate in both species. Electrophysiology studies will clarify if this is the case.

The tuning curves of the P-EN and P-EG neurons have the same statistics because in our models we assumed that all neurons have the same biophysical properties and since both these types of neurons receive the same inputs their responses are identical.

## Connectivity differences affect response dynamics

Despite the substantial similarity in functional structure of the two circuits, the subtle differences in connectivity affected the dynamics of the circuit behaviour. This became apparent when we compared the response of both circuits to sudden changes of heading (*Figure 7*). At a qualitative level, the fruit fly heading signal (the 'bump') could jump abruptly from one state to another, whereas the locust circuit exhibited a gradual transition. The results obtained with our spiking neuron models were corroborated by rate-based implementations of the models (*Figure 7—figure supplement 1*), confirming that the observed difference in response dynamics is not a consequence of neuron model choice but rather due to the differences in connectivity.

To explore whether this difference in movement dynamics of the heading signal could be a result of the different inhibition patterns produced by the Delta7 neurons, we replaced the global Delta7 connectivity pattern in the fruit fly model with the connectivity pattern of the locust Delta7 neurons, effectively swapping the fruit fly version of these cells with the locust version. Both the Delta7 to Delta7 and the E-PG to Delta7 connections were replaced with that of the locust. The data generated by this hybrid-species model revealed that changing the global inhibition to local inhibition was sufficient to produce the gradual 'bump' transition we observed in the locust circuit (*Figure 8*).

**Table 2.** Characteristics of the neuron tuning curves.

The Full Width at Half Maximum (FWHM), the peak impulse rate of each family of neurons and the activity amplitude measured as the range of firing rates are shown. Numbers are given as median and standard deviation. The activity of Delta7 neurons in *Drosophila* is approximately even, hence the corresponding FWHM measurement is not meaningful and marked as 'N/A'.

| Neuron class | *Drosophila* | | | Locust | | |
|---|---|---|---|---|---|---|
| | FWHM | Peak | Amplitude | FWHM | Peak | Amplitude |
| | (°) | (imp./s) | (imp./s) | (°) | (imp./s) | (imp./s) |
| E-PG | 94.7 ± 4.0 | 208.4 ± 2.3 | 208.2 ± 2.2 | 73.4 ± 2.6 | 220.8 ± 1.4 | 220.8 ± 1.4 |
| P-EN | 74.6 ± 3.8 | 230.3 ± 2.3 | 230.3 ± 2.3 | 58.9 ± 3.1 | 163.6 ± 0.9 | 163.6 ± 0.9 |
| P-EG | 74.6 ± 3.8 | 230.3 ± 2.3 | 230.3 ± 2.3 | 58.9 ± 3.1 | 163.6 ± 0.9 | 163.6 ± 0.9 |
| Delta7 | N/A | 289.9 ± 1.8 | 58.1 ± 4.2 | 96.0 ± 3.2 | 265.4 ± 2.9 | 265.4 ± 2.9 |

## Quantification of the ring attractor responsiveness

Having shown that small changes in the morphology of the Delta7 cells affect the dynamics of the heading signal in a qualitative way, we next quantified the maximal rate of change each ring attractor circuit could attain. To this end, we measured the time it took for the heading signal to transition from one stable location to a new one, in response to different angular heading changes of the stimulus. This was carried out for all three models: the fruit fly model, the locust model, and the hybrid-species model. The fruit fly ring attractor circuit stabilised to the new heading in approximately half the time it takes for the locust circuit to stabilise, across different magnitudes of angular heading change (*Figure 8A*). The hybrid-species circuit had a similar response time to the locust circuit. This confirmed that the pattern of inhibition in the network is the main contributor to the observed effect.

To calculate the maximal rate of angular change each circuit can possibly track we divided the angular heading change by the time required for the heading signal to transition. When moving gradually, the heading signal transitions along the shortest path around the ring attractor. Therefore, in the calculation of the angular change rate, the numerator was the shortest angular distance between the two azimuths, calculated as

$$angle = \begin{cases} angle, & \text{if } angle \leq 180° \\ 360° - angle, & \text{if } angle > 180° \end{cases} \tag{1}$$

The resulting angular rate of change values revealed that the circuit found in the fruit fly is significantly faster than the locust circuit and the hybrid-species circuit with localised inhibition (*Figure 8B*). The rate of change was maximal for angular displacement of 180°, because this is the maximum azimuth distance the bump has to travel, as for all other angular displacements there is a shorter path.

## Effects of varying the uniformity of inhibition

The above results strongly suggested that the different pattern of inhibition is instrumental to generating the different dynamics in the two circuits. Up to this point, we have examined two extreme cases of inhibitory synaptic patterns, that of the global, uniform, inhibition found in *Drosophila* and the localised inhibition found in the locust. However, in principle, there could be any degree of uniformity of the inhibition between these two extremes. So far, the locust inhibition has been modelled as a summation of two Gaussian functions that approximates the synaptic density across the PB glomeruli, as derived from estimates of dendritic density along the PB in dye-filled Delta7 neurons (*Heinze and Homberg, 2007*; *Beetz et al., 2015*; *Hadeln et al., 2020*). In the fruit fly, the synaptic distribution of Delta7 neurons has been modelled as uniform across the PB glomeruli, although there might be subtle synaptic density variation along their length. To account for this possibility, we explored a range of synaptic terminal domains distributions. As no measurements of synaptic strengths exist for either animal, we asked what effect varying the synaptic terminal distribution would have on the ring attractor behaviour. We thus modelled the inhibitory synaptic strength across the PB using two Gaussian functions, with peaks separated by 7 or 8 glomeruli, and varied their width (standard deviation σ, see also section Materials and methods). This would not only give us the effect of different inhibitory synaptic domain widths but also predict the plausible range of widths that the actual animals must have in order to exhibit the observed dynamics.

Modelling these variations showed that the transition mode of the heading signal depended on both the extent of the inhibitory synaptic domain width and the angular heading change of the stimulus. This sets limits on the plausible standard deviation (σ) range that the synaptic strength distribution must obey in the actual animals (*Figure 9*). We observed that for both circuits there was a range of low σ values, corresponding to more localised inhibition, which produce gradual transitions ('locust-like'). As σ was increased, the inhibitory pattern became more uniform or global, and both circuits transitioned to abrupt jumps ('fly-like'). Based on density estimates of dendrites in the PB, we approximated the inhibitory synaptic distribution with a value of $\sigma = 0.8$ for the locust model, yielding a gradual activity transition regime across the whole range of angular changes. These results suggested that the pattern of inhibition is indeed key to the circuit dynamics in response to rapid heading changes.

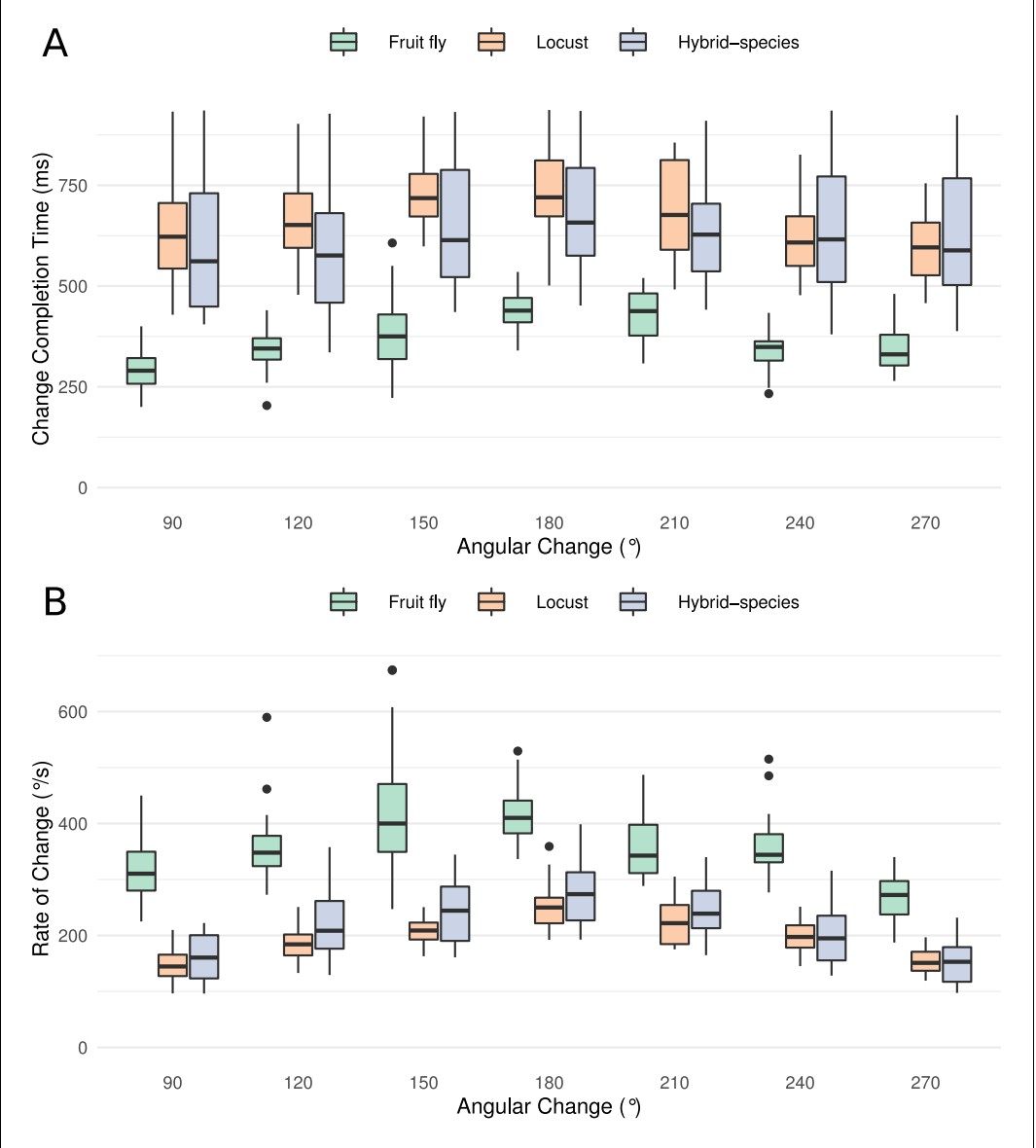

**Figure 8.** Transition time and rate of the heading signal. (**A**) Time required from the onset of the stimulus until the heading signal settles to its new state. The abscissa (horizontal axis) displays the azimuthal difference between initial and target azimuth. (**B**) The maximum rate of angular change each model can attain computed as the ratio of shortest angular change of stimulus divided by transition duration. The values for different magnitudes of heading change are depicted as medians. The boxes indicate the 25th and 75th percentiles while the whiskers indicate the minimum and maximum value in the data after removal of the outliers (black dots). 'Hybrid-species' is the combination of the fruit fly model with the locust inhibition pattern.

However, the morphology of the Delta7 neurons is not the only difference between the ring attractors in the two species, hence the recorded response patterns are not identical for the two species (*Figure 9*). There is also anatomical difference in the presence of the P-EN to E-PG feedback loops only in the locust and consequently the synaptic efficacies differ between the two models. We investigate the effect of this anatomical difference in the subsequent section.

## Attractor states distribution

We next investigated the attractor basin of each model. The finite size of the two circuits renders them discrete approximations of ring attractors (*Brody et al., 2003*). As a consequence, in the

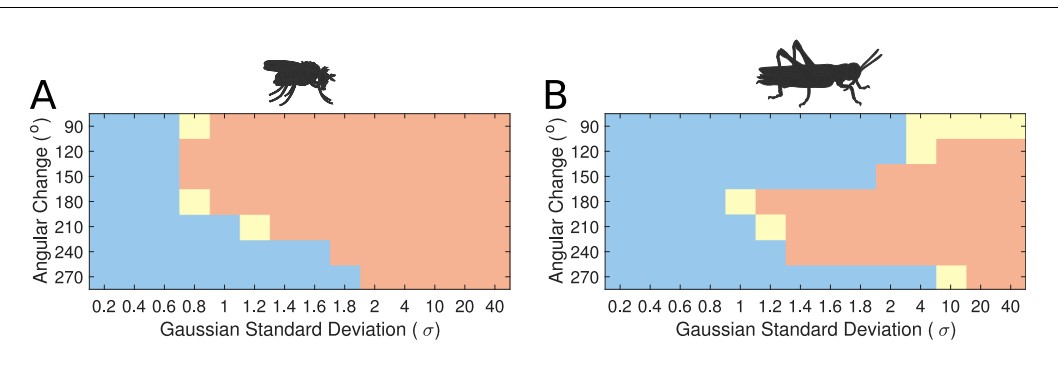

**Figure 9.** Transition regime as function of inhibitory uniformity. Heading signal transition regime for (**A**) the fruit fly ring attractor circuit and (**B**) the desert locust ring attractor circuit. Blue denotes gradual transition of the heading signal, orange denotes abrupt transition (jump), and yellow marks trials that were producing both gradual and abrupt transitions (for definitions see section Materials and methods).

absence of input, the activity 'bump' will tend to settle to one of a discrete set of states (note this does not prevent a continuous encoding of heading while a heading stimulus is provided, which could be decoded by downstream neuronal circuits). We tested this by stimulating the E-PG neurons at varying azimuthal locations around the circuits, then removing the input stimulus and examining the position of the activity 'bump' after 3 s. Both the fruit fly and locust circuits had discrete attractor states where the heading signal eventually settled once the stimulus was removed. Typically, the heading signal moved to the nearest attractor state. When a stimulus was applied equidistantly between two attractor states then, once the stimulus was removed, the activity 'bump' moved to one of the two attractor states stochastically due to the presence of noise in the system (*Figure 10*). These attractor states were more stable and clearly delineated in the locust while in the fruit fly there was a wider distribution of 'bump' locations, indicating that the locust ring attractor is more robust to drift and noise (*Figure 10*).

## Stability characteristics of the ring attractors
### The locust head direction circuit is more robust to noise
An important aspect of a ring attractor is its stability characteristics. The differences in the distribution of activity 'bump' locations reported in the previous section hinted that the locust ring attractor is more robust to noise. To quantify this property of the two ring attractors, we measured the effect of different levels of structural (synaptic) noise to the circuit stability. The ring attractor of the locust was significantly more tolerant to structural noise than the fruit fly circuit (*Figure 11A*).

However, these two ring attractors differ in several respects. To identify the reason for the reduced sensitivity of the locust model to synaptic noise we compared the locust with the hybrid-species model. These two models differ in that reciprocal connections between P-EN and E-PG neurons are present only in the locust model (*Figure 3* and *Figure 4*). If these reciprocal connections are responsible for the increased robustness of the circuit, we would expect the locust model to be more robust to synaptic noise than the hybrid-species model. This is exactly what we found (*Figure 11A*), thus we inferred that these reciprocal connections, between P-EN and E-PG neurons, provide the increased robustness to the locust model. This circuit specialisation might have important repercussions to the behavioural repertoire of the species, enabling locusts to maintain their heading for longer stretches of time than fruit flies, an important competence for a migratory species such as the locust.

### P-EG neurons stabilise the head direction circuit
In our models, we included the P-EG neurons connecting the PB glomeruli with EB tiles. Unlike the P-ENs, these neurons have the same connectivity pattern as the E-PG neurons but with presynaptic and postsynaptic terminals on opposite ends. What is the effect of the P-EG neurons in the circuit? Effectively, the P-EG neurons form secondary positive feedback loops within each octant of the

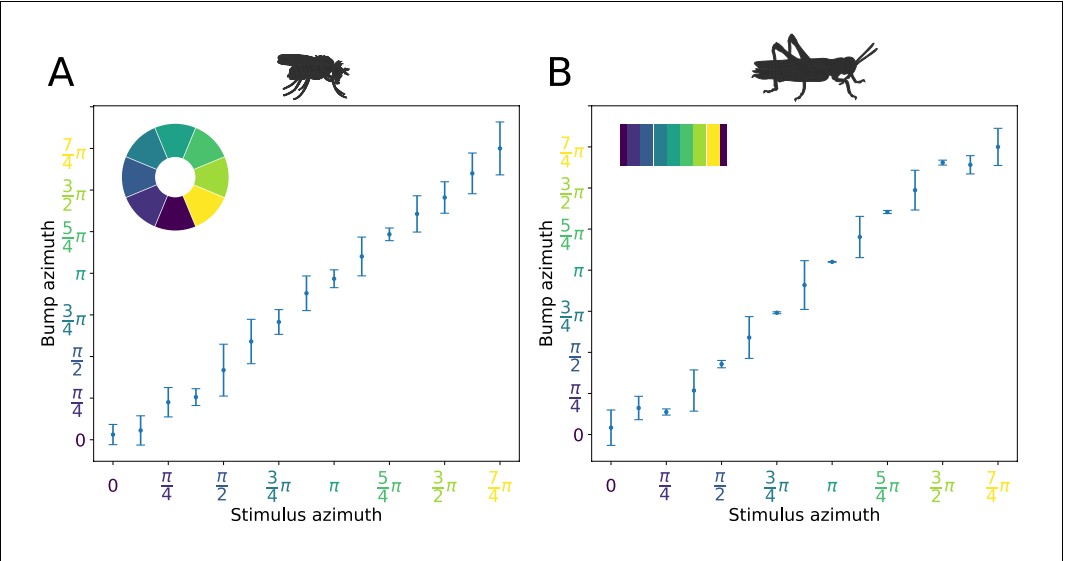

**Figure 10.** Distribution of activity 'bump' locations. The distribution of azimuthal location of the heading signal 3 s after stimulus removal is plotted. On the abscissa (horizontal axis), the azimuth where the stimulus is applied is shown. On the ordinate (vertical axis), the mean location and standard deviation of the activity 'bump' azimuth, 3 s after the stimulus is removed, are shown. (**A**) for the fruit fly and (**B**) for the locust. Inset images depict the corresponding EB tiles in colour. Smaller standard deviation corresponds to the 'bump' settling more frequently to the same azimuth. This is the case when the stimulus is applied near an attractor state. Applying the stimulus equidistantly from two attractor states results in a movement of the 'bump' to either of them and hence the increased standard deviation. In the locust when stimulating the ring attractor at one of the attractor states the 'bump' tends to settle at it, indicated by the reduced standard deviation at these locations. In the fruit fly, the activity 'bump' is prone to noise and not as stable, thus the standard deviation is not as modulated. This means that the locust attractor states are more stable resulting to the smaller dispersion of 'bump' location.

circuit that, we hypothesised, help the heading signal to be maintained stably in the current position, even when lacking external input. Therefore, we expected the circuit to function as a ring attractor without these connections, but to be more vulnerable to drift if the neuronal connection weights are not perfectly balanced. The recurrent P-EG to E-PG loops should counteract this tendency to drift.

We tested this hypothesis by measuring the effect of imposing imbalance in the connectivity strengths of P-EN to E-PG neurons between the two hemispheres. We did this for both the full fruit fly and locust circuits as well as two altered circuits with the P-EG neurons removed. The synaptic strengths for the four circuits were optimised separately, since completely removing the P-EG neurons without appropriate synaptic strength adjustment breaks the ring attractor. We measured the percentage of simulation runs that resulted in a stable heading signal being maintained for at least 3 s. The presence of the P-EG neurons substantially increased the robustness of both species models to the effects of synaptic strength imbalance in the P-EN to E-PG synapses, as a stable heading signal was observed over a far wider range of synaptic efficacy changes (*Figure 11B&C*). The P-EG neurons therefore contribute significantly to the tolerance of the ring attractors to synaptic strength asymmetries.

## Effect of inhibition to stability

It is interesting to note that even though in the locust model the reciprocal connections between E-PG and P-EG neurons were weaker than in the fruit fly model, the presence of the extra reciprocal connections between P-EN and E-PG neurons in the locust resulted in a more stable ring attractor than that in the fruit fly, which possesses only one but stronger recurrency loop. Finally, the hybrid-species model was more robust than the fruit fly one (*Figure 11A*). The fruit fly and the hybrid-species models differed in the width of their inhibitory synaptic domains and in their synaptic strengths. Although their difference in robustness was smaller than the previously examined ones, we can see an effect of the inhibitory pattern on the stability of the circuit.

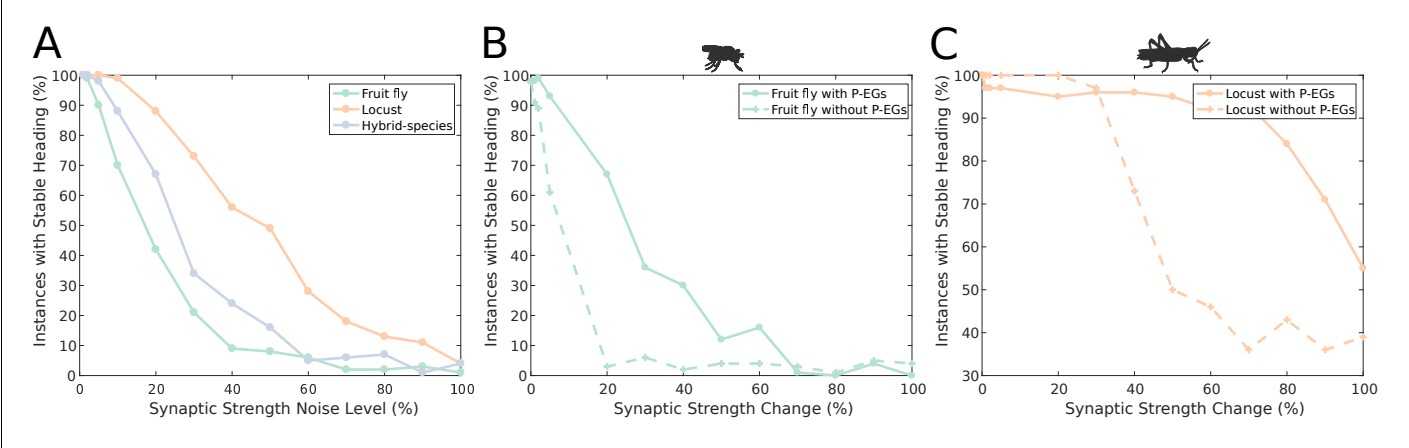

**Figure 11.** Effect of synaptic efficacy heterogeneity on ring attractor stability. (**A**) Stability of the ring attractor heading signal for the fruit fly, locust and hybrid-species (fruit fly with localised inhibition) model as a function of heterogeneity in the synaptic efficacies across all synapse types (modelled as additive white Gaussian noise). (**B, C**) Stability of the ring attractor heading signal as a function of structural asymmetry introduced by deviating synaptic efficacies between P-EN and E-PG neurons when the circuit includes the P-EG neurons versus when they are removed. In all three plots, the percentage of trials that result in a stable activity 'bump' is shown. On the horizontal axis the absolute value of percentile synaptic strength change is shown. Number of trials n = 100 for each level of noise. With P-EG neurons both ring attractors are more tolerant to such structural asymmetries. The locust ring attractor is more robust to both types of structural noise.

## Effect of neuronal heterogeneity

Until this point, we have assumed that all neurons have identical properties. We now relax this assumption by making the membrane properties of the neurons heterogeneous. We tested the effect of neuronal heterogeneity to the stability of the ring attractors. Overall, the stability of the ring attractors deteriorated with increased deviation from the nominal values of membrane properties (*Figure 12*), but the locust model was more robust to these membrane property variations. Importantly, the distinct heading signal transition regimes (gradual transition in the locust model versus jump in the fruit fly model) were preserved regardless of heterogeneous membrane properties across the neuronal population (*Figure 12—figure supplement 1*).

## Response to proprioceptive stimuli

Mechanistically, Turner-Evans et al. showed that the activity of P-EN neurons in one hemisphere of the brain increases when the animal turns contralaterally, both with and without visual input (*Turner-Evans et al., 2017*). The increase in activity is related to the angular velocity the fly experiences (*Turner-Evans et al., 2017*). Whereas the origin of the angular velocity information in darkness is not known, efference copies of motor commands or proprioceptive inputs are the most likely sources of information about the fly's rotational velocity. To test whether our models reproduce this behaviour, we artificially stimulated the P-EN neurons in one hemisphere of the PB, mimicking an angular velocity signal caused by turning of the animal, and observed the effect on the heading signal (*Figure 13*).

Both the locust and the fruit fly model reproduced the response dynamics reported by *Turner-Evans et al., 2017*. Exploration of the response of the circuit to different stimulation strengths showed that the rate by which the heading signal shifts around the ring attractor increases exponentially with increase of uni-hemispheric stimulation strength (*Figure 14*). While this general relationship was consistent between the two species, the increase was much steeper in the fly. Additionally, the required stimulus for initiating 'bump' shifting was lower in the fruit fly. Both of these aspects concur with the faster response rate of the fruit fly model to positional stimuli and support their ability to track fast body saccades even when only angular velocity input is available.

Continuous application of angular velocity input caused the heading signal to reach an edge of the PB and then wrap around and continue on the other edge. This behaviour is present in both models and is thus independent of the physical shape of the EB, that is, whether it forms a closed ring or possesses open ends. The wrapping around of the heading signal is required for the animals

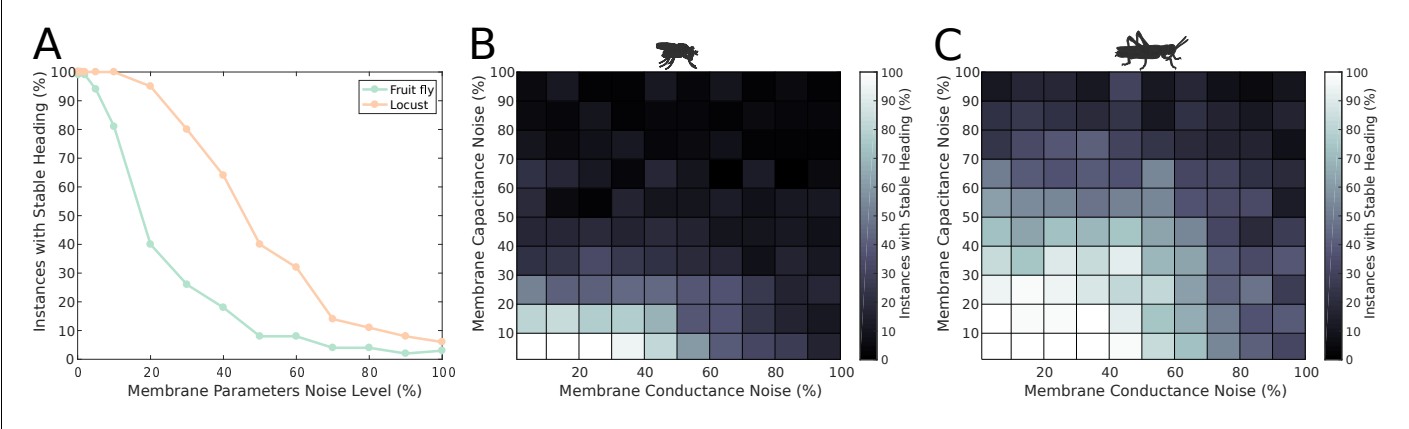

**Figure 12.** Effect of membrane parameter heterogeneity on ring attractor stability. (**A**) Stability of the ring attractor heading signal for the fruit fly and the locust model when the membrane properties are heterogeneous across the neuronal population. (**B, C**) Stability of the ring attractor heading signal when the level of noise on conductance and capacitance is varied independently. In all three plots, the percentage of trials that result in a stable activity 'bump' as a function of heterogeneity in cell membrane properties is shown (number of trials n = 50 for each condition). The locust ring attractor is more robust to white Gaussian noise in both conductance and capacitance. In both cases, the activity 'bump' is more tolerant to conductance variation than capacitance.

The online version of this article includes the following figure supplement(s) for figure 12:

**Figure supplement 1.** Effect of cell membrane parameter heterogeneity to transition regime.

to track movements that involve turning around its body axis for more that 360° and is supported by the effective closed ring structure we found in both species.

## Discussion

The availability of tools for the study of insect brains at the single neuron level has opened the way to deciphering the neuronal organisation and principles of the underlying circuit's behaviour.

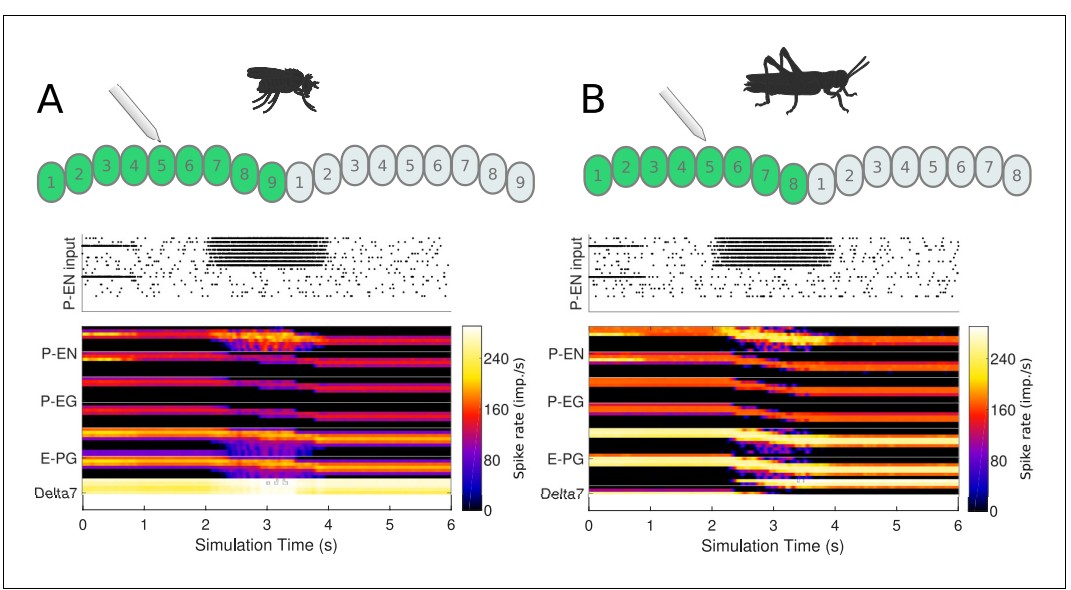

**Figure 13.** Response to uni-hemispheric stimulation. Upper plots show the P-EN stimulation protocol and corresponding induced P-EN activity; lower plots show the response of the ring attractor for (**A**) the fruit fly circuit and (**B**) the locust circuit. The initial bilateral stimulation initialises a persistent activity 'bump', which moves around the circuit in response to stimulation of P-ENs in all the columns in one hemisphere only.

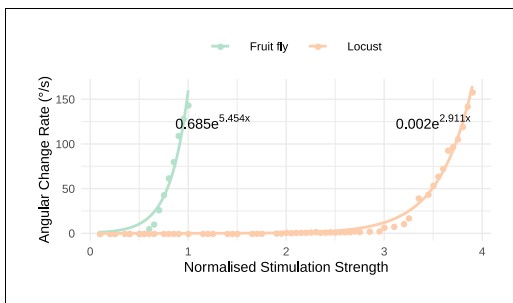

**Figure 14.** Response to uni-hemispheric stimulation. Response rate of change of the heading signal with uni-hemispheric stimulation of P-EN neurons. The angular rate of change increases exponentially with stimulation strength and does so most rapidly for the fruit fly circuit. The data points have been fit with the function $y = ae^{bx}$ and the parameters of the fitted curves are shown on the plot.

However, even where there is progress towards a complete connectome, the lack of data on synaptic strengths, neurotransmitter identity, neuronal conductances, etc. leave many parameters of the circuit unspecified. Exploring these parameters via computational modelling can help to illuminate the functional significance of identified neural elements. We have applied this approach to gain greater insight into the nature of the heading encoding circuit in the insect central complex (CX), including the consequences of differences in circuit connectivity across two insect species.

## Overall conservation of structure and function

We have focused on a subset of neurons in the PB and EB which have been hypothesised to operate as a ring attractor, with a 'bump' of neuronal activity moving across columns consistently with the changing heading direction of the animal. The neuronal projection patterns and columnar organisation differ between the two insect species we have analysed, the fruit fly and the locust. There are additional morphological columns in the PB of flies (9 vs. 8), resulting in a different number of functional units that could influence the symmetry of the underlying neural circuits. Also, the EB in the fruit fly forms a physical ring, while the homologous region in the locust is an open structure. Our analysis of the connectivity as a directed graph has revealed, surprisingly, that the circuits are nevertheless equivalent in their effective structure, forming a closed ring attractor in both species with an identical functional role for each neuron class. The preservation of this circuit across 400 million years of evolutionary divergence suggests that it is an essential, potentially fundamental, part of the insect brain.

It is worth noting that an essential part of the circuit, namely the functionally closed ring that we found in both species, is achieved with two different solutions. In the fruit fly, the torus-shaped EB provides an anatomical solution to the closure of the ring via overlapping projections from E-PG neurons innervating the innermost and outermost PB glomeruli. In contrast, in the locust the midline spanning output fibers of the E-PG neurons in the medial PB glomeruli serve this function in combination with a slightly different projection pattern that results in the P-EN neurons forming reciprocal connections back to E-PG neurons in the same octant. In this context, it is interesting to note that neither solution to this problem is possible for insects of a different order, the lepidoptera (moths and butterflies). These insects have an almost straight EB, their PB is split along the midline, and right-left connections between the two halves are realised by a neuropil-free fiber bundle (*Heinze and Reppert, 2012*; *Adden et al., 2020*). Neither midline crossing E-PG fibers within the PB, nor local connections around the ring of the EB are therefore morphologically possible, suggesting that a functional closure of the heading direction circuit is either not required or achieved via other means in these species. The notion that there are many solutions to the same problem is further highlighted by data from bumblebees showing the existence of a ninth E-PG neuron that connects the medialmost PB glomerulus to the outermost ipsilateral EB wedge, closing the ring in yet another way (personal observations, S.H.). Exploring these different solutions across many species could provide key insights into the evolution of this circuit under a multitude of evolutionary history constraints.

In combination, these findings underline that the large-scale anatomical differences at the level of neuropils and projection patterns do not necessarily affect the core functions of the circuit. Rather, the functional constraints appear significant enough that even in those parts of the circuit that clearly differ between species convergent solutions have evolved that solve similar problems, albeit in slightly different ways.

## Differences in dynamical response

Surprisingly, more subtle differences in the morphology between the two species have significant effects on the dynamical response of the heading direction circuit. First, the shape of the dendritic arborizations of one type of CX neuron determines how quickly the model circuit tracks rotational movements. Second, a difference in the overlap of neuronal projections in the EB results in an extra feedback loop between the P-EN and E-PG neurons in the locust circuit that makes it more robust to synaptic noise.

We suggest that the effects of these differences are consistent with the behavioural ecology of the two species. On the one hand, the faster response of the ring attractor circuit in the fruit fly accommodates the fast body saccades that fruit flies are known to perform (*Tammero and Dickinson, 2002*; *Fry et al., 2003*). On the other hand, the locust is a migratory species, so its behaviour demands maintenance of a defined heading for a long period of time (*Homberg, 2015*; *de Vries et al., 2017*). This requirement for heading stability might have provided the selective pressure needed to drive the evolution of a more noise resilient head direction circuit.

## Assumptions and simplifications

As any model, our circuits are necessarily condensed and simplified versions of the real circuits in the insect brain. In comparison to previous models, the work we present has been more precisely constrained by the latest anatomical evidence. We additionally constrained our models to use plausible values for the biophysical properties of neurons (membrane conductance and capacitance) as well as spiking rates (background activity) supported by electrophysiological evidence. Furthermore, in building our models we did not assume that the underlying circuits must be ring attractors, but rather asked and investigated whether, given the available connectivity data, they can be. This was especially the case for the locust model since our work represents the first model of this circuit to date. Nevertheless, it is important to outline those areas where our assumptions cannot be fully justified from the existing data and identify the potential consequences for the modelling results.

### Morphological assumptions

In our model of the fruit fly heading tracking circuit, we assumed a uniform distribution of dendrites across the Delta7 neurons. Imaging of these neurons suggests that there might be a subtle variation of the dendritic density along their length. However, it is unclear how this subtle variation might be related to synaptic density and efficacy. We, therefore, initially made the simplifying assumption that these neurons have uniform synaptic efficacy across the PB. However, we also explored the effect of varying the degree of synaptic uniformity, showing that there is a range of synaptic efficacy distributions that still can produce the fly-like rapidity in the circuit response.

In general, arborization trees of neurons in the CX can be very complex, as they are not only confined to specific slices, but also to one or several layers, especially within the EB. In *Drosophila*, the spiny terminal arbors of E-PG neurons extend to the width of single wedges in the EB, occupying both the posterior and medial layers. In contrast, P-EG and P-EN neurons arborize in tiles, hence innervating only the posterior surface volume of the EB (*Wolff and Rubin, 2018*). Therefore, we assume that presynaptic terminals of P-EG and P-EN neurons form synapses with E-PG postsynaptic terminals in the posterior layer of the EB. In locusts, the E-PG arborizations are more complex, as these cells innervate a single wedge of the anterior and medial EB layers, but extend at least twice this width to either side in the posterior layer that provides overlap with the P-EN neurons (*Heinze and Homberg, 2008*). Additionally, the wider fibres have a different morphological appearance. P-EG neurons in this species innervate all layers evenly. Although these detailed differences likely have consequences for connectivity, we simplified these arborizations to their most essential components, aiding the extraction of the core features. With the advance of comparative connectomics, these aspects will become accessible for investigation.

### Connectivity assumptions

Several assumptions were made while deriving the neuronal connectivity in our models. We assumed well delineated borders of synaptic domains, which is clearly not always the case. Especially in the EB, some overlapping of neighbouring synaptic domains due to stray terminals is to be expected (*Wolff et al., 2015*). The circumferential extent of arbors in wedges and tiles may affect the integrity

of the resulting circuit and its properties. However, due to lack of adequate data about the extent of such overlap we cannot currently model this aspect in a sensible way.

Furthermore, neuronal connectivity was mostly inferred by co-location of neuronal arbors, that is, projection patterns. A functional connectivity study has reported that stimulation of E-PG neurons triggered significant responses to Delta7 neurons but no columnar neurons (*Franconville et al., 2018*). However, as those authors note, the lack of response might be due to the limitations of the method used. Alternatively, such connections might be mediated by interneurons instead of being monosynaptic. Future work using electron microscopy data will elucidate which of the overlapping arborizations correspond to functional connections and allow us to augment our models.

## Functional assumptions

Further assumptions were made about neuronal polarity, type of synapses and synaptic efficacies. *Lin et al., 2013* characterise the EB arbor of E-PG neurons in *Drosophila* as having both presynaptic and postsynaptic domains; however, *Wolff et al., 2015* report that using anti-synaptotagamin is inconclusive for presynaptic terminals. In our models for both the fruit fly and the locust we thus assumed that E-PG neurons are purely postsynaptic in the EB, following the most parsimonious polarity estimate. Connectomics data from a recent preprint demonstrate that in *Drosophila* synapses exist that directly link Delta7 to E-PG neurons in the PB (*Turner-Evans et al., 2019*). These synapses are most likely inhibitory and would thus inhibit the distal portion of the ring and thus would not alter the location of the activity 'bump'. For simplicity and because they do not affect the functional layout of the circuit, these synapses were not included in our model.

Furthermore, the Delta7 neurons are assumed to have inhibitory effect on their postsynaptic neurons, as *Kakaria and de Bivort, 2017* proposed. However, there is evidence that Delta7 neurons make both inhibitory and excitatory synapses to other neurons (*Franconville et al., 2018*). Indeed, these cells were recently shown to be glutamatergic, enabling both inhibitory and excitatory effects on postsynaptic cells via different glutamate receptors (*Turner-Evans et al., 2019*). As the downstream neurons with demonstrated excitatory responses (P-FN neurons) are not part of our current model, we made the simplifying assumption that Delta7 neurons have exclusively inhibitory effect on their postsynaptic neurons, both in flies and locusts. It is also possible that there are other sources of inhibition in the circuit, for example mediated by the GABAergic ring neurons originating in the bulbs, as suggested by *Green and Maimon, 2018*, or via GABAergic Gall-EB ring neurons (*Turner-Evans et al., 2019*). We do not explore this possibility in our current work.

We additionally assumed that the synaptic strengths of all synapses of each class are identical. This might not be the case in the actual animals, especially considering that one of the EB tiles (T1) is innervated by twice as many neurons as other tiles in fruit flies (*Figure 3*). Neurons innervating this tile might have reduced synaptic efficacy in order to maintain the radial symmetry of the circuit intact. Similarly, the synaptic strengths of the neurons closing the ring in locusts would be expected to be different than those of other synapses if the ring does not have a functional 'seam'. Such a synaptic efficacy variation is suggested by the fact that the arborization density of E-PG neurons innervating the two medial PB glomeruli (G9 and G1) is not the same in both of them. There is certainly space for further exploration of the effect of synaptic efficacy in those segments of the ring in both species. Finally, synaptic strength variation might exist for the two Delta7 neurons that have presynaptic terminals in three glomeruli instead of two (*Table 3*).

## Biophysical assumptions

All types of neurons in our models were assumed to have the same nominal biophysical properties even though anatomical evidence has shown that their morphology, somata size and main neurite thickness differ (*Heinze and Homberg, 2008*). To relax this assumption, we explored the effect of heterogeneity in the biophysical properties of the neuronal population. We corroborated our conclusions using both rate-based and Leaky Integrate and Fire neurons with refractory period. This allowed us to highlight the significance of the neuronal connectivity on the circuit dynamics. The point spiking neuron model was sufficient for investigating the performance characteristics, spike timing dynamics and potential spike synchronisation effects in the ring attractors when exposed to neuronal noise, but clearly is highly abstracted in comparison to real neurons. However, we lack most of the necessary detail to constrain more complex neural models. One caveat is that intrinsic

properties of neurons could provide short-term memory that would radically alter the circuit response. It is not possible to explore this possibility with the models we have used, but we can conclude that such properties do not appear to be necessary for generating basic ring attractor dynamics. Furthermore, it will be interesting to study how differences in the biophysical properties of neurons between the two species might be affecting performance. We are not exploring this possibility here.

## Comparison to 'canonical' ring attractor models

In our work, we compared the hypothetical heading tracking circuit of two evolutionary distant species. We went beyond mere simulation of neuronal projection data by analysing and deriving the effective underlying circuit structure of the two ring attractors. Our analysis and derivation of the complete effective neuronal circuits revealed not only differences in dynamics but also the construction principles of these circuits. This approach allowed us to identify elements that differ in several ways from the 'canonical' ring attractor described in earlier theoretical models (e.g. *Amari, 1977*; *Skaggs et al., 1995*; *Zhang, 1996*).

For example, the circuit found in the two insect species combines two functionalities in the P-EN neurons that are typically assigned to separate neural populations in computational models of ring attractors. Such computational models use one set of neurons to provide the lateral excitation to nearest neighbours and a different set of neurons that receive angular velocity input to drive the left-right rotation of the heading signal. In the insect circuit, the P-EN cells are part of the lateral excitation circuit, providing excitation to their two nearest neighbours, but they also receive angular velocity input. This difference is suggestive of a more efficient use of neuronal resources than the typical computational models of ring attractors. Another novel element we found in the insect ring attractors is the presence of local feedback loops within each octant of the circuit structure (P-EG to E-PG and P-EN to E-PG). Both of these feedback loops increase the tolerance of the ring attractors to noise.

## Hypotheses regarding circuit differences

Another unique aspect of our modelling work is the comparison of related, but not identical, circuits found in two species. Indeed, using computational modelling allows us to investigate 'hybrid' circuits, combining features of each, in order to try to understand the functional significance of each observed difference independently. Nevertheless, some differences between these circuits are not explained by the current model, and may require additional work to fully explicate.

One question is what is the role, if any, of the ninth PB glomeruli found so far only in *Drosophila*? In particular, the existence of the innermost glomeruli that are not innervated by the P-EN neurons seems perplexing. The same signals from tile 1 of the EB are sent to both ends of each hemisphere of the PB (glomeruli 1 and 9) and from there action potentials propagate along the Delta7 neurons along the PB length. Our speculation is that this may be a mechanism to reduce the distance and time these signals have to travel to cover the full PB, that is, the maximum distance any signal must travel is only half of the distance it would need to propagate from one end of the PB to the other as in other species, such as the locust. If this is the case, it would constitute one more specialisation in *Drosophila* that reduces the response time of the ring attractor. It therefore seems that several specialisations have been orchestrated in minimising the response delays in fruit flies. Testing this idea would require multi-compartmental models to capture the action potential transmission time along neurites; as argued above, this may be contingent upon first obtaining detailed biophysical characterisation of the Delta7 neurons.

Another remaining question is what is the role of the closed ring-shaped EB in *D. melanogaster*. One possibility is that such a closed ring topology would allow local reciprocal connections between P-EN and E-PG neurons all around the EB ring, as reported in *Turner-Evans et al., 2019*. This would allow direct propagation of signals between these neurons within the EB instead of requiring them to travel via the PB, as in the current model, again increasing the speed with which the heading direction can be tracked and allowing smoother transition between neighbouring tiles. Note that such direct reciprocal connections within the EB can only span the full ring with a closed ring anatomy and would not be possible between the two ends of the EB in the locust. To investigate the potential effect of such hypothetical reciprocal connections within the EB, further studies are

required. Possibly blocking signal transmission via the PB to isolate functional connectivity within the EB would allow comparison of signal transmission time measurements within the EB versus via the PB. Such measurements would determine how different and hence significant those two pathways might be in the ring attractor performance.

A further hypothesis relates to the evolutionary lineage of these two features in the *Drosophila* CX. It will be of interest to study whether the ring shaped EB appeared before or after the appearance of the ninth glomeruli. One possibility is that the EB evolved into a ring shape after the appearance of the ninth glomeruli in the PB, allowing connections from one common tile to both glomeruli 1 and 9 and hence providing such a common driving signal. Alternatively, a pre-existing ring-shaped EB might have allowed the evolution of usable ninth glomeruli that resulted in faster propagation. Similarly, the P-EN to E-PG recurrency found only in the locust might be an acquired adaptation of the locust that increases robustness to noise, or an ancestral feature that has been lost in fruit flies.

Comparison of different species could potentially elucidate such questions. We would expect individual species to have a selective subset of the specialisations we found, endowing them with brain circuits supporting the behavioural repertoire suiting their ecological niche. It will, therefore, be informative to analyse the effective heading direction circuit of other species, spanning evolutionary history, in order to get insights into how such adaptations relate to and accommodate behaviour. Our results emphasise the importance of comparative studies if we are to derive general principles about neuronal processing, even in systems that appear highly conserved such as the CX head direction circuit in insects. Many of the circuit properties observed in *Drosophila* appear to reflect specific evolutionary adaptations related to tracking rapid flight manoeuvres. Despite the many strengths of *Drosophila* as an experimental model, it therefore remains important to ground conclusions about the insect brain in comparison with other species.

## Materials and methods

### Neuron model

Our models used the source code of *Kakaria and de Bivort, 2017* as a starting point. We used Leaky Integrate and Fire neuron models with refractory period (*Stein, 1967*). The membrane potential of each neuron was modelled by the differential equation

$$\frac{dV_i}{dt} = \frac{1}{C_m}\left(\frac{V_0 - V_i}{R_m} + I_i + \sum_{j=1}^{N} M_{j,i}I_j\right) \tag{2}$$

where $V_i$ is the membrane potential of neuron $i$, $V_0$ the resting potential, $R_m$ the membrane resistance, $C_m$ the membrane capacitance, $I_i$ the external input current of neuron $i$, $M_{j,i}$ the network connectivity matrix, $I_j$ the output current of each neuron in the circuit and $N$ is the number of neurons.

The model parameter values including membrane resistance, capacitance, resting potential, undershoot potential and postsynaptic current magnitude ($I_{PSC}$) were set to the same values as used by *Kakaria and de Bivort, 2017*. These values are consistent with evidence from measurements in *D. melanogaster* and other species. The membrane capacitance $C_m$ is set to $2nF$ and the membrane resistance $R_m$ to $10M\Omega$ for all neurons, assuming a surface area of $10^{-3}cm^2$ (*Gouwens and Wilson,*

**Table 3.** Neuronal nomenclature.

The names used for the homologous neurons differ between *Drosophila* and other species. The first column shows the name used in this paper to refer to each group of neurons. The other three columns provide the names used in the literature.

| Model | *Drosophila* | | Locust |
|---|---|---|---|
| Neuron name | Consensus name | Systematic name (*Wolff and Rubin, 2018*) | Name |
| E-PG | E-PG and E-PG$_T$ | PBG1-8.b-EBw.s-D/V GA.b and PBG9.b-EB.P.s-GA-t.b | CL1a |
| P-EN | P-EN | PBG2-9.s-EBt.b-NO1.b | CL2 |
| P-EG | P-EG | PBG1-9.s-EBt.b-D/V GA.b | CL1b |
| Delta7 | Delta7 or $\Delta$7 | PB18.s-Gx$\Delta$7Gy.b and PB18.s-9i1i8c.b | TB1 |

*2009*). The resting potential $V_0$ is set to $-52mV$ for all neurons (*Rohrbough and Broadie, 2002*; *Sheeba et al., 2008*). The action potential threshold is $-45mV$ (*Gouwens and Wilson, 2009*). When the membrane potential reaches the threshold voltage an action potential template is inserted in the recorded time series. No other impulses occur during this period operating in effect as a refractory period. The action potential template is defined as *Kakaria and de Bivort, 2017*:

$$V(t) = \begin{cases} V_{thr} + (V_{max} - V_{thr})\frac{\mathcal{N}\left(\frac{t_{tp}}{2}, \left(\frac{t_{AP}}{2}\right)^2\right) - \alpha_1}{\beta_1}, & \text{if } 0 \leq t < \frac{t_{AP}}{2} \\ V_{min} + (V_{max} - V_{min})\frac{sin\left(\left(t - \frac{t_{AP}}{2}\right)\frac{2\pi}{t_{AP}} + \frac{\pi}{2}\right) + \gamma_1}{\delta_1}, & \text{if } \frac{t_{AP}}{2} \leq t \leq t_{AP} \end{cases} \tag{3}$$

where $V_{max}$ is the peak voltage set to $20mV$ (*Rohrbough and Broadie, 2002*). $V_{min}$ is the action potential undershoot voltage, set to $-72mV$ (*Nagel et al., 2015*). $t_{AP}$ is the duration of the action potential set to 2 ms (*Gouwens and Wilson, 2009*; *Gaudry et al., 2013*). $\mathcal{N}(\mu, \sigma^2)$ is a Gaussian function with a mean μ and standard deviation σ. $\alpha_1$, $\beta_1$, $\gamma_1$, and $\delta_1$ are normalisation parameters for scaling the range of the Gaussian and the sinusoidal to 0 to 1.

The firing of an action potential also adds a postsynaptic current template to the current time series. The postsynaptic current template is defined as

$$I(t) = \begin{cases} I_{PSC}\frac{sin\left(\frac{t\pi}{2} - \frac{\pi}{2}\right) + \alpha_2}{\beta_2}, & \text{if } 0 \leq t < 2ms \\ I_{PSC}\frac{2^{-(t-2)/t_{PSC}} + \gamma_2}{\delta_2}, & \text{if } 2ms \leq t \leq 2ms + 7t_{PSC} \end{cases} \tag{4}$$

where $I_{PSC} = 5nA$ (*Gaudry et al., 2013*). Excitatory and inhibitory postsynaptic currents are assumed to have the same magnitude but opposite signs. $t_{PSC} = 5ms$ is the half-life of the postsynaptic current decay (*Gaudry et al., 2013*). $\alpha_2$, $\beta_2$, $\gamma_2$, and $\delta_2$ are normalisation constants so that the range of the sinusoidal and exponential terms is 0 to 1. The postsynaptic current traces have duration $2ms + 7t_{PSC}$, that is $2ms$ of rise time plus $7t_{PSC}$ of decay time. The simulation was implemented using Euler's method with a simulation time step of $10^{-4}s$. Our simulation code is derived from the source code published by *Kakaria and de Bivort, 2017*. All simulations were performed using MATLAB (The MathWorks Inc, Natick, MA) and all source codes are available at https://github.com/johnpi/eLife_Pisokas_Heinze_Webb_2019 (copy archived at https://github.com/elifesciences-publications/eLife_Pisokas_Heinze_Webb_2019; *Pisokas, 2020*). For data analysis we used MATLAB, python, and R scripts.

## Neuronal projections and connectivity

We modelled and compared the hypothetical ring attractor circuits of the fruit fly *D. melanogaster* and the desert locust *S. gregaria*. The connectivity of the circuits has been inferred mostly from anatomical data derived using light microscopy, with overlapping neuronal terminals assumed to form synapses between them (*Wolff and Rubin, 2018*; *Wolff et al., 2015*; *Heinze and Homberg, 2007*; *Heinze and Homberg, 2008*; *Pfeiffer and Homberg, 2014*).

Our models include the E-PG, P-EG, P-EN and Delta7 neurons. Note that, in fruit flies, P-EG refers to the updated set of neurons innervating all PB glomeruli as reported in *Wolff and Rubin, 2018* (PBG1-9.s-EBt.b-D/V GA.b). In this paper, E-PG refers to the E-PG (PBG1-8.b-EBw.s-D/V GA. b) and the complimentary E-PG$_T$ (PBG9.b-EB.P.s-GA-t.b) combined (*Wolff et al., 2015*; *Wolff and Rubin, 2018*). Therefore, E-PG neurons are innervating all PB glomeruli in both species. Delta7 refers to PB18.s-Gx$\Delta$7Gy.b and PB18.s-9i1i8c.b neurons combined (*Wolff et al., 2015*; *Wolff and Rubin, 2018*). *Table 3* shows the nomenclature correspondence in detail.

These neurons innervate two of the central complex neuropils, the protocerebral bridge (PB) and the ellipsoid body (EB). Ellipsoid body is the name used for this structure in the fruit fly *D. melanogaster*, while in the locust *S. gregaria* the equivalent structure is referred to as lower division of the central body (CBL). To aid comparisons with previous models and for general simplification, we use the term EB for both species. The PB is a moustache shaped structure consisting of 16 or 18 glomeruli, depending on the species. In the fruit fly *D. melanogaster*, the EB has a torus shape consisting of eight tiles. Each tile is further broken down in two wedges. In the locust *S. gregaria*, the EB (CBL) is a linear structure, open at the edges, consisting of eight columns. Each column has two subsections similar to the wedges found in *D. melanogaster*.

For both *D. melanogaster* and *S. gregaria*, the synaptic domains of each of the E-PG, P-EN and P-EG neurons are confined to one glomerulus of the PB, with the exception of the locust E-PG neurons that cross-innervate the two medial glomeruli (*Figure 3* and *Figure 4*). In the EB, the synaptic domains of E-PG neurons are constrained in single wedges (half tiles) while the synaptic domains of P-EN and P-EG neurons extend to whole tiles (*Wolff et al., 2015*). Furthermore, E-PG neurons innervate wedges filling the posterior and medial shells of the EB while P-EG neurons innervate whole tiles filling only the posterior shell of the EB (*Wolff et al., 2015*). Our model assumes that their overlap in the posterior shell implies functional connectivity.

In our models, the E-PG, P-EG and P-EN neurons are assumed to produce excitatory effect on their postsynaptic neurons while Delta7 neurons are assumed to provide the inhibition, as *Kakaria and de Bivort, 2017* proposed. The projection patterns of the aforementioned neurons were mapped to one connectivity matrix for each species (*Figure 1—figure supplement 1*). *Figure 1—figure supplement 1A* shows the connectivity matrix of the *Drosophila melanogaster* fruit fly model, *Figure 1—figure supplement 1B* the connectivity matrix of the *S. gregaria* desert locust model.

The most salient difference between the two matrices is the connectivity pattern of the Delta7 neurons (lower right part of *Figure 1—figure supplement 1A* and *Figure 1—figure supplement 1B*). In *D. melanogaster*, the Delta7 neurons receive synapses uniformly across the PB glomeruli, while in the locust *S. gregaria* the Delta7 neurons have synaptic domains focused in specific glomeruli. We analysed the effect of this difference in detail in the Results section. Another major difference apparent in the connectivity matrices is the existence of 18 glomeruli in the PB of *D. melanogaster* but 16 in *S. gregaria*.

We modelled each PB glomerulus, as being innervated by one neuron of each class (E-PG, P-EG, P-EN) even though in reality there are several instances of each one. This was done in order to simplify the computational demands of the simulations.

The locust inhibition pattern has been modelled as the summation of two Gaussian functions that approximate the synaptic density across the PB glomeruli, as derived from estimates of dendritic density along the PB in dye-filled Delta7 neurons. The standard deviation (σ) of the Gaussian functions was set to the value 0.8 as the nearest approximation to the visually determined synaptic domain width. To calculate the synaptic strength of each synapse we used the expression

$$w(i) = W \frac{1}{\sigma\sqrt{2\pi}} e^{-\frac{1}{2}\left(\frac{\frac{i-1}{n}2\pi-\mu}{\sigma}\right)^2} \tag{5}$$

where $W$ is a scaling factor specifying the maximum synaptic strength across the PB, $i$ is the glomerulus number as shown in *Figure 15*, $n$ is the number of glomeruli in each hemisphere, $\mu = \pi$, and σ is the standard deviation parameter specifying the width of the Gaussian function used. σ is the parameter estimated by visual inspection of light microscopy data. $W$ is the parameter selected by the optimisation process.

It is worth noting that in all our simulations we use the full connectivity matrices derived from neuronal projection data and not the effective circuits described in the section Results.

## Stimuli

Two types of input stimuli were used for the experiments: heading and angular velocity. The heading stimulus was provided as incoming spiking activity directly to the E-PG neurons, corresponding to input from Ring neurons (*Young and Armstrong, 2010*) (called TL neurons in locusts [*Vitzthum et al., 2002*]). The position of a visual cue, angle of light polarisation (*Heinze and Homberg, 2007*) or retinotopic landmark position (*Seelig and Jayaraman, 2015*) around the animal, was mapped to higher firing rates supplied to E-PG neurons at the corresponding location of the EB. The stimulus followed spatially a von Mises distribution with mean the azimuth of the stimulus and full width at half maximum (FWHM) of approximately 90° (*Figure 16*). The spatial distribution of the stimulus strength was derived using *Equation 6*.

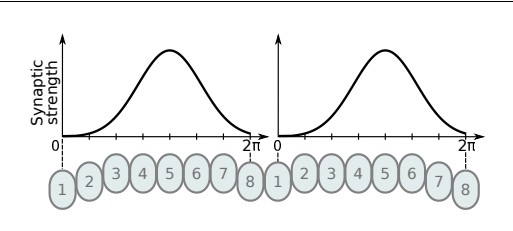

**Figure 15.** Illustration of Gaussian distribution of synaptic strengths. The Gaussian distribution of synaptic strengths along synapses located in the PB glomeruli. The synaptic strengths along the PB are illustrated for one Delta7 neuron. The example illustrates the distribution for eight glomeruli, the same method is used for the hybrid-species model using nine glomeruli instead.

$$f(\mu, x) = \frac{e^{\kappa cos(a(x)-\mu)}}{2\pi I_0(\kappa)},$$

$$I_0(\kappa) = \sum_{i=0}^{\infty} \frac{\kappa^{2i}}{2^{2i}(i!)^2}, \qquad (6)$$

$$a(x) = \frac{\pi}{4}(x-1)$$

where $\mu \in [0, 2\pi]$ is the stimulus centre location parameter, $x = \{1, 2, \ldots, 8\}$ is the EB tile numerical index and $\kappa = \frac{3}{4}\pi$ is the shape parameter. The values returned by $f(\mu, x)$ are converted to corresponding spiking activity levels. To do this, we sampled from a Poisson distribution. The minimum value is mapped to the background activity level and the maximum to the peak level of activity. We assumed that the background activity follows a Poisson distribution with a mean background action potential rate of 5 impulses/s. The peak impulse firing rate of the stimulus signal was equal to the peak spiking rate of the activity 'bump' across the E-PG neuron population under steady state conditions, in order to obtain comparable measurements across species.

The second type of stimulus, angular velocity stimulus, consisted of spikes which were directly supplied to all P-EN neurons in one hemisphere of the PB, corresponding to the direction of rotation (clockwise versus counter-clockwise). The peak impulse rate of the injected spike trains was equal to the peak rate of the steady state activity 'bump' across the P-EN neurons. This was done in order to allow for direct comparisons between species.

## Free parameters

The free parameters of our models are the synaptic efficacies. The efficacies of synapses connecting each class of neurons are assumed to be identical, e.g., all P-EN to E-PG synapses have the same strength. Therefore, we have one free parameter for each synaptic class. Furthermore, we reduced the computational complexity of optimising the synaptic strengths by making the synaptic strength between some classes of neurons identical. The synaptic strengths of E-PG to P-EN and P-EG are identical as are the synaptic strengths of Delta7 to P-EN and P-EG. This is the minimum set of synaptic strengths that results in working ring attractors. We assumed that all synapses are excitatory apart from the synapses with Delta7 neurons on the presynaptic side, which were assumed to be inhibitory, as *Kakaria and de Bivort, 2017* proposed. The synaptic strength was modelled as the number of $I_{PSC}$ unit equivalents flowing to the postsynaptic neuron per action potential.

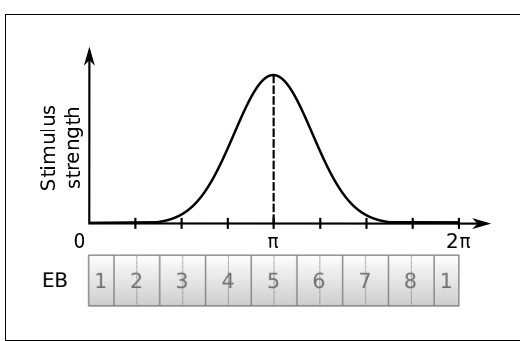

**Figure 16.** Illustration of von Mises distributed stimulus. The curve demonstrates the relative intensity of the stimulus supplied to neurons innervating each EB tile. In this illustration the stimulus is centred at tile 5.

Although our models are constrained by anatomical evidence, existing biological studies do not specify synaptic weights or connectivity. Based on the goal that each of the circuits should yield a functional ring attractor, an optimisation algorithm was used to search for synaptic strength combinations that resulted in working ring attractors. Both simulated annealing and particle swarm optimisation algorithms were used (Matlab Optimization Toolbox 'simulannealbnd' and 'particleswarm' functions); the first one converges quicker while the second one covers the search space more thoroughly. We constrained the acceptable solutions to those that produced an activity 'bump' with full width at half maximum (FWHM) of approximately 90° since this is the

width that has been observed in fruit flies (*Kim et al., 2017*).

The objective function used to optimise the synaptic strengths $w_i$ was:

$$\underset{\mathbf{w}}{\mathrm{argmin}} \quad 4(\epsilon_{H1}(\mathbf{w}) \quad + \epsilon_{H2}(\mathbf{w})) + \epsilon_{W1}(\mathbf{w}) + \epsilon_{W2}(\mathbf{w}) + Np_0(\mathbf{w})$$

$$
\begin{aligned}
\text{s.t} \quad \epsilon_{H1}(\mathbf{w}) &= \frac{|H_d(t_1) - H_a(\mathbf{w}, t_1)|}{360°} \\
\epsilon_{H2}(\mathbf{w}) &= \frac{|H_d(t_2) - H_a(\mathbf{w}, t_2)|}{360°} \\
\epsilon_{W1}(\mathbf{w}) &= \frac{|90° - W_a(\mathbf{w}, t_1)|}{360°} \\
\epsilon_{W2}(\mathbf{w}) &= \frac{|90° - W_a(\mathbf{w}, t_2)|}{360°} \\
p_0(\mathbf{w}) &= \frac{1}{N}\sum_{i=1}^{N}(e^{-|w_i|})^2 \\
0 &\leq w_1 \leq 100 \\
0 &\leq w_2 \leq 100 \\
0 &\leq w_3 \leq 100 \\
-100 &\leq w_4 \leq 0 \\
-100 &\leq w_5 \leq 0
\end{aligned}
\tag{7}
$$

where $\epsilon_{H1}$, $\epsilon_{H2}$, $\epsilon_{W1}$ and $\epsilon_{W2}$ are the error factors measured as deviations from the desired values. $p_0$ is used to penalise synaptic strengths being too close to 0. $N$ is the number of synaptic strengths $w_i$. $H_d(t)$ is the desired activity 'bump' heading at time $t$, while $H_a(\mathbf{w}, t)$ is the actual measured activity 'bump' heading at time $t$ given a model with synaptic strengths $\mathbf{w}$. $W_a(\mathbf{w}, \mathbf{t})$ is the actual measured width of the activity 'bump' at time $t$ (measured as the full width at half maximum). The constraints in *Equation 7* specify that the synapses with Delta7 neurons at their presynaptic side are inhibitory (negative) and all others are excitatory (positive). Synaptic weights were initialised with values $-0.01$ or $0.01$ depending on whether the negative only or positive only constraint was applied. During optimisation the spiking models were used to run the simulations and search the space of synaptic strengths. The synaptic strength sets that resulted from multiple runs were manually tested to verify the results. The objective function was used to optimise the synaptic strengths separately for each of the models: the fruit fly, the locust, and the hybrid-species model.

## Sensitivity analysis and parameter noise

For the sensitivity analysis, white Gaussian noise was added to the membrane parameters of neurons (conductance and capacitance) as well as to the synaptic efficacies, using the formula

$$
\begin{aligned}
v_i &= v_{nominal} + \frac{x}{100} v_{nominal}\epsilon, \\
\epsilon &\sim \mathcal{N}(\mu, \sigma^2)
\end{aligned}
\tag{8}
$$

where $v_i$ is the resulting noisy value of the parameter with $i = \{1, 2, \ldots, M\}$ and $M$ being the number of parameters. $v_{nominal}$ is the nominal value of the parameter, $x \in [0, 100]$ is the percentage of noise to be added to the nominal value, $\varepsilon$ is a random variable sampled from the Gaussian distribution with $\mu = 0$ and $\sigma^2 = 1$. When noise was added to the conductance and capacitance of neurons the resulting values were clipped to a minimum of 0 because conductance and capacitance values cannot be negative. For measuring the tolerance to inter-hemispheric synaptic asymmetry we altered the P-EN to E-PG synapses in one hemisphere by different amounts in the range $-100\%$ to $100\%$.

The number of successful trials was counted in each condition. The criterion for a successful trial was that the activity 'bump' transitioned from an initial stimulus-driven heading to a second stimulus-driven heading with an error of less that $\pm 45°$ and subsequently the second heading was maintained for at least 3 s. The criterion used for judging jump versus gradual transition of the heading signal was that for the transition to be considered a jump the intervening neurons between the origin and end location must not become maximally active during the transition.

## Acknowledgements

The authors thank Matthias Hennig and Michael Rule for their invaluable comments.

# Additional information

## Funding

| Funder | Grant reference number | Author |
|---|---|---|
| H2020 European Research Council | Grant agreement no. 714599 | Stanley Heinze |
| University Of Edinburgh | Graduate Student Fellowship | Ioannis Pisokas |

The funders had no role in study design, data collection and interpretation, or the decision to submit the work for publication.

## Author contributions

Ioannis Pisokas, Conceptualization, Software, Formal analysis, Validation, Investigation, Visualization, Methodology, Writing - original draft, Writing - review and editing; Stanley Heinze, Conceptualization, Supervision, Funding acquisition, Validation, Visualization, Writing - review and editing; Barbara Webb, Conceptualization, Resources, Supervision, Funding acquisition, Writing - review and editing

## Author ORCIDs

Ioannis Pisokas https://orcid.org/0000-0001-7426-3207
Stanley Heinze https://orcid.org/0000-0002-8145-3348

## Decision letter and Author response

Decision letter https://doi.org/10.7554/eLife.53985.sa1
Author response https://doi.org/10.7554/eLife.53985.sa2

# Additional files

## Supplementary files

• Transparent reporting form

## Data availability

All source scripts for producing the data as well as for generating Figures 6A, 6B, 7, 8, 9, 10, 11, 12, 14 and Tables 1 and 2 are located at https://github.com/johnpi/eLife_Pisokas_Heinze_Webb_2019 (copy archived at https://github.com/elifesciences-publications/eLife_Pisokas_Heinze_Webb_2019).

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
