## [Decision Letter]

**Acceptance summary:**

This study disentangles navigational circuitry in the central complex of locusts, showing that the effective connectivity exhibits a ring-like structure that encodes head direction. Importantly, such circuit structure resembles that of another insect species, *Drosophila*, with a key difference in the profile of recurrent and inhibitory connections. Simulations show how these differences in connectivity may navigate a tradeoff between the stability and responsiveness of head direction encoding that is tuned to the motor behaviour of each species.

**Decision letter after peer review:**

Thank you for submitting your article "The head direction circuit of two insect species" for consideration by *eLife*. Your article has been reviewed by three peer reviewers, one of whom is a member of our Board of Reviewing Editors, and the evaluation has been overseen by Ronald Calabrese as the Senior Editor. The following individual involved in review of your submission has agreed to reveal their identity: Benjamin L de Bivort (Reviewer #3).

The reviewers have discussed the reviews with one another and the Reviewing Editor has drafted this decision to help you prepare a revised submission.

Summary:

This study uses existing quantitative neuroanatomy data in *Drosophila* and locust to construct a connectivity matrix of the presumptive ring attractor circuit used in navigation. Using a simplification of the connectivity the authors show a “hidden” ring attractor in the locust that has a distinct connectivity profile from *Drosophila*. Using simulations the authors examine the consequences of the different connectivity in each species and conclude that the locust has a more robust internal representation of heading, with possible ethological consequences.

Essential revisions:

1) Reviewers were concerned about a lack of detail and possible ambiguity in how the reduced/effective connectivity diagrams are obtained. The authors should revise the diagrams and accompanying text, providing additional explanatory figures if necessary.

2) Conclusions are based on the contribution of connectivity to dynamics alone, yet the choice of modelling detail is inconsistent. The authors should show that the most parsimonious assumptions (e.g. rate-based models) corroborate the results and include robustness analysis of the more “detailed” simulations. Statements about the biological realism should be appropriately edited.

*Reviewer #1:*

Summary:

This work uses existing quantitative neuroanatomy data in *Drosophila* and locust to construct a connectivity matrix of the presumptive ring attractor circuit used in navigation. They show using an elegant topological analysis that there is a “hidden” ring attractor in the locust that has a distinct connectivity profile. Using integrate and fire based simulations the authors examine the consequences of the different connectivity in each species and conclude that the locust has a more robust internal representation of heading, with possible ethological consequences. The manuscript is beautifully presented and highly relevant, however there are unfortunate and gravely misleading statements about the assumptions in the model that need to be addressed. A final, positive attribute of the work is an important broad message that cautions against drawing overly general conclusions from the details of a heavily studied model system, something the community urgently needs to be reminded of. If the substantive concerns can be addressed the manuscript could make a valuable contribution to systems/circuit neuroscience.

Major comments:

1) The conclusions of this study rest on the assumption that all of the neuronal properties in both circuits are identical and conform to simple integrate and fire dynamics with a linear membrane. This is almost certainly false but little to no mention is made of the “potential consequences” of this assumption. No attempt has been made to test the sensitivity of the conclusions to this assumption, even though this is possible without resorting to “multicompartmental models”. The only acknowledgement of this is made in the Discussion which simply asserts that this is essentially a technical simulation detail that would not add “tangible gains”. Worse still, the authors claim elsewhere that the model “uses realistic biophysical properties”. The point extends beyond heterogeneity and it is a little painful to have to remind the authors of how false their statement is; the writing indicates they are almost certainly aware of this. Intrinsic properties alone can provide short term memory that would radically alter the responsiveness of a circuit and there are thousands of neurophysiological studies that make this very clear. The manuscript could have been written in a more scientifically honest and circumspect way, pointing out these issues without necessarily exhaustively addressing all of them. Instead, unfortunately, the authors have chosen to mislead their readers. This makes the manuscript unpublishable in its present form.

2) The optimization algorithm and the results obtained from it are inadequately described and validated. There are many ways of using particle swarm optimisation, for example, and there are initial values/distributions/search limits that affect the results. None of this is described. Making code available is not a substitute.

3) The choice of detail in the modelling is a little bizarre. On one hand, it is clear that this is an abstract model, placing synaptic connectivity above all else. This would suggest a rate based model. Instead, integrate and fire models are used that have additional parameters as well as a highly detailed spike waveform inserted in place of a spike event. This is a cosmetic feature that only obscures the results. Do any of the conclusions depend on these decisions, and if not, why were these decisions made? Again, this decision unfortunately looks like an attempt to convince the reader of biophysical realism in a trivial and inconsequential way, while ducking the substantive issues.

4) I suggest the authors build up their modelling approach from a truly minimal (e.g. rate based) case, proceeding to details like spiking and proper heterogeneity. This would allow readers to get some sense of which details “may” actually matter. The manuscript should also be more scientifically honest about the limitations of the study because these are necessary given available data and tractability. The manuscript would actually be stronger if this were made clear.

*Reviewer #2:*

The authors simulated models based on the circuit's structure of the CX of the fruit fly and of the locust while optimizing free parameters, and demonstrated a yielded trade-off in functionality between the two models: faster respond to changes in fruit fly circuit vs higher tolerance to noise in locust circuit.

The scientific significance of the work may be debated, but I find the manuscript unclear.

Figure 1:

Panels G-H are too small, which makes them almost unreadable.

Make horizontal and vertical lines to separate the different groups (P-EN/P-EG/E-PG/Delta7)

Number the ticks within each group to indicate the order of the glomeruli (might have been either 1-2-3-….-9-1-2-…-9 or 1-1-2-2-3-3….9-9).

The gray/black bars that indicate borders between groups in panel 1G seem to be a bit off (between P-EN and P-EG, and between E-PG and Delta7 -both in x and y axes).

It is unclear what the extra two empty rows/columns at Delta7 mean. Maybe I am missing something, so please explain it explicitly in the figure legend.

There seem be several mistakes in the connectivity description as given in panels 1G-H, it is hard to be sure given the notation of the matrices (order of glomeruli) are unclear.

Figure 3:

It is unclear what additional information panel D provides given panel A, and similarly what additional information panel E provides given panel B.

In panel F, P-EG are not numbered.

Although in practice there is no difference between the pattern of connectivity from E-PG to P-EN and the pattern of connectivity from E-PG to P-EG, in principle there could have been. Therefore I suggest that the two patterns will be indicated in two separate panels.

I understand the attempt for simplicity, nevertheless I suggest making the illustration in panel F much larger, and separate each note in the graph into two (the two glomeruli of each ordinal number). I believe that the 8-fold symmetry can be still appreciated by the reader, on top of allowing a clearer description of the circuit connectivity. (It is perfectly fine to exclude in this illustration the 9th glomerulus for simplicity, and indicate the pattern of its connections in a supplementary figure, as currently done).

Figure 4:

Same comments as given for Figure 3.

Maybe I am missing something about the notations but there seem to be several discrepancies between panels A-E and panel F. For example: given panel A there should be an arrow from yellow node 5 to green node 6, given panel B there should not be an arrow from green node 5 to yellow node 4, given panel C there should be an arrow from orange node (5) to yellow node 6, and given panel D there should be an arrow from yellow node 6 to orange node (5).

Figure 5:

The illustration is not accurate since each E-PG is shown to send its output only to one Delta7. In case it is too hard to illustrate the connectivity form E-PG to Delta7, authors may exclude E-PG from this illustration altogether. Additionally, as only the connections from Delta7 1 to other Delta7 are shown in black and the connections from the rest of Delta7 are shown in gray -I suggest making also only the connections from Delta7 1 to the relevant P-EG and P-EN in black and the connections from the rest of Delta7 in gray (for consistency).

Figure 10:

The mean bump azimuth for stimulus 7/4 π seem not to 7/4 pi. Is it indeed the case? If yes, then why?

Overall, the bump azimuth seems to be biased toward smaller angle than stimulus azimuth (and given that, I am surprised not to see any cases of bump azimuth 7/4 π for stimulus azimuth 0).

"We next investigated the fixed points of each ring attractor… circuits had eight discrete attractor states", in this case these are not ring attractors.

*Reviewer #3:*

In this manuscript, Pisokas et al. consider a neural circuit that is conserved across insect species: the reciprocal connections between the protocerebral bridge and ellipsoid body. There is evidence this circuit supports goal-directed navigation in several species. Using neuroanatomical data from locust and fruit flies, the authors simulate the dynamics of this circuit in each species. There are three major conclusions. First, the connectivity of the locust circuit has the potential to implement a ring-attractor. Second, the simulated dynamics in the locust model reiterate the qualitative physiological results consistent with a ring-attractor. Third, in a comparative analysis, the authors determine that differences between the locust and fly circuits (different presynaptic extents in inhibitory PB neurons and reciprocal connections between P-ENs and E-PGs in the locust) render the locust spatial dynamics more stable, but also slower to respond to stimulus changes. The authors speculate that these differences may relate to differences in the ethology of the species. The comparative circuit dynamics approach is innovative and interesting, and this may be an early example of a technique that becomes increasingly valuable and prevalent as more connectomic data is produced in non-model species.

The authors do a persuasive job of justifying these conclusions. The conclusions are relatively mild in their ambitions, and there are ways to restructure this approach that might offer more in terms of substantive conclusions and less of a feeling of a rather ad hoc, piecewise analysis. For example, the authors could delineate ~6 circuits: fly, locust, hybrid, locust without PEGs, fly without PEGs, hybrid without PEGs, as the rows of a matrix, in which the columns are analyses, bump movement speed, rate of change, number of locations, stability etc. If that's too much for a table, then the core figures could have the same panel arrangement of the same ~6 (or how many ever) circuit configurations. This is just an idea, and might be outside the scope of revisions because the authors have achieved their stated goal of those three main conclusions.

As for presentation, the figures are generally clear. The text is a bit wordy, but it makes it more palateable to get through all the neuroanatomical jargon. There are a fair number of typos/grammatical issues as indicated below. We almost certainly did not catch them all, so the authors will have to do careful editing. Because the writing is not super dense, the typos did not really interfere with understanding.

Note: This review was done by a π and grad student in the same lab, so it's long because it’s the consensus of both reviews. We did not find any major concerns that would call into question the major conclusions.

More substantive concerns:

– Clearer definitions of ring attractor would be helpful early on. What are the requirements in neuroanatomy for a circuit to have the structure of a ring-attractor. What are the requirements in physiological dynamics and stimulus response for the behavior of a circuit to be consistent with a ring attractor? Operational definitions would be particularly valuable because the authors frame their investigation as a test of the existence of such properties in the circuitry.

– More needs to be said in the Results section about the synapse strength parameterization. What was the cost function for this optimization, etc. Even in the Materials and methods section, this is covered rather briefly.

– The hybrid model is good, but the authors might consider titrating the number of glomeruli with postsynaptic sites in the Delta7 neurons from 3 to 7, and show the response in stimulus shift + other "ring attractor" behaviors. This is not necessary to the conclusions they present, but would be interesting from a theoretical perspective and there may be other species with morphologies like that.

– Unclear why 9th glomerulus in fly is okay to exclude from Figure 3.

– We wondered if the wraparound of the 8th tile to the 1st tile in locust is justified. If E-PG5 innervates tiles T5 and T6 and glomeruli G5 of the PB, would E-PG8 innervate the right-most wedges of Figure 4, i.e. T8 and T1? Then, as P-EN5 innervates tiles T5 and T6, does P-EN8 innervate T8 and both half-tiles of T1 on opposite sides of the crescent?

– Please make clear how the effective circuits are different from the full connectivity matrices, is the former simplified to give intuition? Are there reasons for reducing the real connectivity matrix to the effective matrix other than computational simplicity?

– Why was the test of the role of P-EGs done by an interhemispheric mismatch? (Also, what does this mean, since the CX is on the midline.) Simply removing that population from the simulation would seem more in line with how other hypotheses were tested.

– The discussion on the closed ring shape of the Dipteran EB confused me. The physical shape doesn't matter much (other than in timing details) if the network connectivity is the same? Why would the closure of the physical ring matter for P-EN to E-PG connectivity, except at the suture, and if it's only there that it might matter, the text needs to say this.

---

## [Author Response]

Essential revisions:1) Reviewers were concerned about a lack of detail and possible ambiguity in how the reduced/effective connectivity diagrams are obtained. The authors should revise the diagrams and accompanying text, providing additional explanatory figures if necessary.

We have clarified the derivation of the ‘effective connectivity’ diagrams with substantially revised Figures 3 and 4 that show step-by-step the reasoning that reveals the effective circuits and have expanded the relevant text accordingly. The revised figures consist of pairs of panels, one showing the neuronal projection pattern and the other the effective connection in a directed graph. We have also clarified that our simulations use the full circuits not the reduced circuits.

2) Conclusions are based on the contribution of connectivity to dynamics alone, yet the choice of modelling detail is inconsistent. The authors should show that the most parsimonious assumptions (e.g. rate-based models) corroborate the results and include robustness analysis of the more “detailed” simulations. Statements about the biological realism should be appropriately edited.

We have confirmed that the key difference between the circuits – sudden activity bump transition versus gradual transition – can be reproduced using rate-based simulations (Figure 7—figure supplement 1), and thus is not dependent on specific assumptions made in the spiking model. We have also expanded the robustness analysis, running simulations with heterogeneous neuronal membrane parameters (membrane conductance and capacitance) as well as random noise on all synaptic weights (updated Figure 11, new Figure 12 and Figure 12—figure supplement 1) and reporting the percentage of trials that produce a bump that follows the stimulus heading as the parameter variance is increased. This shows that our results do not depend on precise tuning of the parameter values or on them having the same value across all units. We have tried to adapt our description of the modelling to be appropriately cautious about the limits on biological realism.

Reviewer #1:Summary:This work uses existing quantitative neuroanatomy data in *Drosophila* and locust to construct a connectivity matrix of the presumptive ring attractor circuit used in navigation. They show using an elegant topological analysis that there is a “hidden” ring attractor in the locust that has a distinct connectivity profile. Using integrate and fire based simulations the authors examine the consequences of the different connectivity in each species and conclude that the locust has a more robust internal representation of heading, with possible ethological consequences. The manuscript is beautifully presented and highly relevant, however there are unfortunate and gravely misleading statements about the assumptions in the model that need to be addressed. A final, positive attribute of the work is an important broad message that cautions against drawing overly general conclusions from the details of a heavily studied model system, something the community urgently needs to be reminded of. If the substantive concerns can be addressed the manuscript could make a valuable contribution to systems/circuit neuroscience.Major comments:1) The conclusions of this study rest on the assumption that all of the neuronal properties in both circuits are identical and conform to simple integrate and fire dynamics with a linear membrane. This is almost certainly false but little to no mention is made of the “potential consequences” of this assumption. No attempt has been made to test the sensitivity of the conclusions to this assumption, even though this is possible without resorting to “multicompartmental models”. The only acknowledgement of this is made in the Discussion which simply asserts that this is essentially a technical simulation detail that would not add “tangible gains”. Worse still, the authors claim elsewhere that the model “uses realistic biophysical properties”. The point extends beyond heterogeneity and it is a little painful to have to remind the authors of how false their statement is; the writing indicates they are almost certainly aware of this. Intrinsic properties alone can provide short term memory that would radically alter the responsiveness of a circuit and there are thousands of neurophysiological studies that make this very clear. The manuscript could have been written in a more scientifically honest and circumspect way, pointing out these issues without necessarily exhaustively addressing all of them. Instead, unfortunately, the authors have chosen to mislead their readers. This makes the manuscript unpublishable in its present form.

It was not our intent to mislead our readers, but we accept that the previous version of the paper did not sufficiently discuss or explore the extent to which the results could be dependent on the choice of integrate and fire units as the basis for the modelling. We have addressed this in several ways in the revision. First, we have confirmed that the key difference between the circuits – sudden activity bump transition versus gradual transition – can be reproduced using rate-based simulations (Figure 7—figure supplement 1), and thus is not dependent on specific assumptions made in the spiking model. Second, we have expanded the robustness analysis, running simulations with heterogeneous neuronal membrane parameters (membrane conductance and capacitance) as well as random noise on all synaptic weights (updated figure 11, new figure 12 and Figure 12—figure supplement 1). This shows that our results do not depend on precise tuning of the parameter values or on them having the same value across all units. We have avoided any use of the term ‘realistic’ and extended our discussion of these issues in the Discussion section ‘Biophysical Assumptions’, including consideration of the possible effect of intrinsic short term neural memory.

2) The optimization algorithm and the results obtained from it are inadequately described and validated. There are many ways of using particle swarm optimisation, for example, and there are initial values/distributions/search limits that affect the results. None of this is described. Making code available is not a substitute.

We apologise for this omission and have now included full details of the algorithms, optimization parameters and objective function used in the Materials and methods.

3) The choice of detail in the modelling is a little bizarre. On one hand, it is clear that this is an abstract model, placing synaptic connectivity above all else. This would suggest a rate based model. Instead, integrate and fire models are used that have additional parameters as well as a highly detailed spike waveform inserted in place of a spike event. This is a cosmetic feature that only obscures the results. Do any of the conclusions depend on these decisions, and if not, why were these decisions made? Again, this decision unfortunately looks like an attempt to convince the reader of biophysical realism in a trivial and inconsequential way, while ducking the substantive issues.

The choice of model detail was motivated principally by the intent to build on the model of Kakaria and de Bivort, 2017, with the explicit inclusion of anatomical connectivity details that differ from their model, but without altering any other details such as the use of a spike waveform (as a convenient technique to not only signify the action potential event but also to produce a refractory period where no further action potentials can be inserted). We chose to use spiking models in order to investigate the possible effects of spike timing interaction given the substantially recurrent structure of the circuits. Rate based models would have obscured any such interactions. The use of spiking neurons is our first step in including more biological detail once it becomes available. We have now verified that the main conclusions still hold for rate-based models, and include discussion of this point in the manuscript.

4) I suggest the authors build up their modelling approach from a truly minimal (e.g. rate based) case, proceeding to details like spiking and proper heterogeneity. This would allow readers to get some sense of which details “may” actually matter. The manuscript should also be more scientifically honest about the limitations of the study because these are necessary given available data and tractability. The manuscript would actually be stronger if this were made clear.

As noted, we have added simulations for both the spiking and rate based implementations of the models for both fruit fly and locust. This illustrates that the difference between sudden activity bump transition versus gradual transition is also holding for the rate based models. These are shown on Figure 7—figure supplement 1, and mentioned at the appropriate part of the manuscript to emphasise that the effects appear to be a property of the connectivity. We have added exploration of the effects of increased heterogeneity in neuronal and synaptic parameters, showing that the conclusions remain robust to significant variation.

Reviewer #2:The authors simulated models based on the circuit's structure of the CX of the fruit fly and of the locust while optimizing free parameters, and demonstrated a yielded trade-off in functionality between the two models: faster respond to changes in fruit fly circuit vs higher tolerance to noise in locust circuit.The scientific significance of the work may be debated, but I find the manuscript unclear.Figure 1:Panels G-H are too small, which makes them almost unreadable.Make horizontal and vertical lines to separate the different groups (P-EN/P-EG/E-PG/Delta7)Number the ticks within each group to indicate the order of the glomeruli (might have been either 1-2-3-….-9-1-2-…-9 or 1-1-2-2-3-3….9-9).The gray/black bars that indicate borders between groups in panel 1G seem to be a bit off (between P-EN and P-EG, and between E-PG and Delta7 -both in x and y axes).It is unclear what the extra two empty rows/columns at Delta7 mean. Maybe I am missing something, so please explain it explicitly in the figure legend.There seem be several mistakes in the connectivity description as given in panels 1G-H, it is hard to be sure given the notation of the matrices (order of glomeruli) are unclear.

We moved the connectivity matrices to a separate supplementary figure to allow them to be magnified and be more clearly visible.

The different groups of neurons are now marked with bars and labels.

Numbering was added to the ticks of the connectivity matrices to indicate the order of neurons.

Bars indicating grouping of neuron types were adjusted to match the borders.

The two empty rows/columns were removed from the connectivity matrices.

The connectivity matrices (panels G and H) were enlarged and moved to a separate supplementary figure to make the details more visible and the added numerical labeling should now aid understanding.

Figure 3:It is unclear what additional information panel D provides given panel A, and similarly what additional information panel E provides given panel B.In panel F, P-EG are not numbered.Although in practice there is no difference between the pattern of connectivity from E-PG to P-EN and the pattern of connectivity from E-PG to P-EG, in principle there could have been. Therefore I suggest that the two patterns will be indicated in two separate panels.I understand the attempt for simplicity, nevertheless I suggest making the illustration in panel F much larger, and separate each note in the graph into two (the two glomeruli of each ordinal number). I believe that the 8-fold symmetry can be still appreciated by the reader, on top of allowing a clearer description of the circuit connectivity. (It is perfectly fine to exclude in this illustration the 9th glomerulus for simplicity, and indicate the pattern of its connections in a supplementary figure, as currently done).

We replaced the figure with an improved version with additional panels explaining the derivation of the effective circuits step by step. This includes separate panels for E-PG and P-EG neurons. Panels D and E were moved to a supplementary figure and replaced with panels showing the connectivity at the edges of the PB which are more helpful for explaining connectivity. Neuron numbering was added in all panels for consistency. All neurons are now numbered, including in panel F. The labeling in the network graphs was updated to more clearly explain what happens to the 9th glomerulus and we labeled the corresponding neurons as “1&9” in the derived circuit to make things clearer for the reader.

Figure 4:Same comments as given for Figure 3.Maybe I am missing something about the notations but there seem to be several discrepancies between panels A-E and panel F. For example: given panel A there should be an arrow from yellow node 5 to green node 6, given panel B there should not be an arrow from green node 5 to yellow node 4, given panel C there should be an arrow from orange node (5) to yellow node 6, and given panel D there should be an arrow from yellow node 6 to orange node (5).

Similarly to Figure 3, we overhauled Figure 4 with an improved version to help explain the derivation of the effective circuits more clearly. Panels were inserted to illustrate the derivation of the effective circuit step by step and panel F is now two panels one with the full circuit and another showing the detail of the ring closing. We believe this supersedes the specific points made above.

Figure 5:The illustration is not accurate since each E-PG is shown to send its output only to one Delta7. In case it is too hard to illustrate the connectivity form E-PG to Delta7, authors may exclude E-PG from this illustration altogether. Additionally, as only the connections from Delta7 1 to other Delta7 are shown in black and the connections from the rest of Delta7 are shown in gray -I suggest making also only the connections from Delta7 1 to the relevant P-EG and P-EN in black and the connections from the rest of Delta7 in gray (for consistency).

Since illustrating accurately the whole connectivity in a single image results in a very cluttered image, we separated the illustration to three images for each animal, which depict the connectivity patterns more accurately.

Figure 10:The mean bump azimuth for stimulus 7/4 π seem not to 7/4 pi. Is it indeed the case? If yes, then why?Overall, the bump azimuth seems to be biased toward smaller angle than stimulus azimuth (and given that, I am surprised not to see any cases of bump azimuth 7/4 π for stimulus azimuth 0).

We fixed plots C and D (now relabeled as A and B), the mapping of tiles to angles along the y axis was indeed incorrect.

"We next investigated the fixed points of each ring attractor… circuits had eight discrete attractor states", in this case these are not ring attractors.

We updated this paragraph to clarify that even though the two circuits structurally form rings, they have a finite number of neurons and thus are discrete approximations of ring attractors. While the circuits are driven by stimulus, the activity bump can smoothly move to any location around the ring and the activity could be decoded by downstream neuronal circuitry as a continuous azimuth value. We therefore use the term “ring attractor” in this sense for simplicity.

Reviewer #3:In this manuscript, Pisokas et al. consider a neural circuit that is conserved across insect species: the reciprocal connections between the protocerebral bridge and ellipsoid body. There is evidence this circuit supports goal-directed navigation in several species. Using neuroanatomical data from locust and fruit flies, the authors simulate the dynamics of this circuit in each species. There are three major conclusions. First, the connectivity of the locust circuit has the potential to implement a ring-attractor. Second, the simulated dynamics in the locust model reiterate the qualitative physiological results consistent with a ring-attractor. Third, in a comparative analysis, the authors determine that differences between the locust and fly circuits (different presynaptic extents in inhibitory PB neurons and reciprocal connections between P-ENs and E-PGs in the locust) render the locust spatial dynamics more stable, but also slower to respond to stimulus changes. The authors speculate that these differences may relate to differences in the ethology of the species. The comparative circuit dynamics approach is innovative and interesting, and this may be an early example of a technique that becomes increasingly valuable and prevalent as more connectomic data is produced in non-model species.The authors do a persuasive job of justifying these conclusions. The conclusions are relatively mild in their ambitions, and there are ways to restructure this approach that might offer more in terms of substantive conclusions and less of a feeling of a rather ad hoc, piecewise analysis. For example, the authors could delineate ~6 circuits: fly, locust, hybrid, locust without PEGs, fly without PEGs, hybrid without PEGs, as the rows of a matrix, in which the columns are analyses, bump movement speed, rate of change, number of locations, stability etc. If that's too much for a table, then the core figures could have the same panel arrangement of the same ~6 (or how many ever) circuit configurations. This is just an idea, and might be outside the scope of revisions because the authors have achieved their stated goal of those three main conclusions.

This is an interesting idea, but we do feel (as the reviewer suggests) that this ‘is outside the scope of revisions’: optimising the parameters and running the simulations for all possible combinations would take significant time without (we believe) significantly adding to the substantive conclusions made in the paper. We have however added some additional combinations (e.g. the locust model with/without PEGs) where this helps to substantiate the core arguments.

As for presentation, the figures are generally clear. The text is a bit wordy, but it makes it more palateable to get through all the neuroanatomical jargon. There are a fair number of typos/grammatical issues as indicated below. We almost certainly did not catch them all, so the authors will have to do careful editing. Because the writing is not super dense, the typos did not really interfere with understanding.

Grammar and language was corrected in multiple places throughout the manuscript. All the changes made can be seen in the submitted document with tracked changes.

Note: This review was done by a π and grad student in the same lab, so it's long because its the consensus of both reviews. We did not find any major concerns that would call into question the major conclusions.

Thank you, we have tried to address all the concerns and comments as detailed below.

More substantive concerns:– Clearer definitions of ring attractor would be helpful early on. What are the requirements in neuroanatomy for a circuit to have the structure of a ring-attractor. What are the requirements in physiological dynamics and stimulus response for the behavior of a circuit to be consistent with a ring attractor? Operational definitions would be particularly valuable because the authors frame their investigation as a test of the existence of such properties in the circuitry.

We added a more explicit outline of the expected dynamics and neuroanatomical features of a ring attractor in the Introduction.

– More needs to be said in the Results section about the synapse strength parameterization. What was the cost function for this optimization, etc. Even in the Materials and methods section, this is covered rather briefly.

We apologise for this omission, we have now included full details of the optimization algorithms, parameters and objective function used in the Materials and methods section.

– The hybrid model is good, but the authors might consider titrating the number of glomeruli with postsynaptic sites in the Delta7 neurons from 3 to 7, and show the response in stimulus shift + other "ring attractor" behaviors. This is not necessary to the conclusions they present, but would be interesting from a theoretical perspective and there may be other species with morphologies like that.

If we have understood the reviewer correctly, this suggestion is essentially equivalent to the analysis we perform in the section ‘Effects of varying the uniformity of inhibition’, where we model the inhibitory synaptic strength across the PB with two Gaussians, and systematically vary the width (σ) (Figure 9). We have tried to clarify this in the text.

– Unclear why 9th glomerulus in fly is okay to exclude from Figure 3.

We have updated Figure 3 and the corresponding text to more clearly explain what happens to the 9th glomerulus, and also label the corresponding neurons as “1&9” in the derived circuit in Figure 3 to make things clearer and assist the reader.

– We wondered if the wraparound of the 8th tile to the 1st tile in locust is justified. If E-PG5 innervates tiles T5 and T6 and glomeruli G5 of the PB, would E-PG8 innervate the right-most wedges of Figure 4, i.e. T8 and T1? Then, as P-EN5 innervates tiles T5 and T6, does P-EN8 innervate T8 and both half-tiles of T1 on opposite sides of the crescent?

We thank the reviewer for bringing to our attention that this aspect was not adequately explained. We have significantly updated Figure 4 and the corresponding text to more clearly explain how the wraparound from the 8th to the 1st tile is justified.

– Please make clear how the effective circuits are different from the full connectivity matrices, is the former simplified to give intuition? Are there reasons for reducing the real connectivity matrix to the effective matrix other than computational simplicity?

The derivation of the effective circuit is performed to gain intuition in the circuit functional structure and to facilitate comparisons between species with different anatomy. Note that for all the results shown in this manuscript we are simulating the whole connectivity matrix not the simplified circuit. We have updated the text to clarify this.

– Why was the test of the role of P-EGs done by an interhemispheric mismatch? (Also, what does this mean, since the CX is on the midline.) Simply removing that population from the simulation would seem more in line with how other hypotheses were tested.

The effect of removing the whole population of P-EG neurons is shown in Figure 11. The interhemispheric mismatch test is an extra test done in addition to the typical white Gaussian noise sensitivity tests to validate our hypothesis that the presence of P-EG neurons renders the circuits more tolerant to synaptic efficacy asymmetries between the two hemispheres which could result in the activity bump drifting and rotating around the ring.

– The discussion on the closed ring shape of the Dipteran EB confused me. The physical shape doesn't matter much (other than in timing details) if the network connectivity is the same? Why would the closure of the physical ring matter for P-EN to E-PG connectivity, except at the suture, and if it's only there that it might matter, the text needs to say this.

As suggested by the reviewer, timing is likely one of the issues that might be different. We updated the relevant paragraph to more clearly explain this. A recent preprint by Vivek Jayaraman’s lab indeed reports the existence of reciprocal connections between P-EN and E-PG cells within the EB of the fruit fly, a connection that is not yet functionally understood, but which is only possible for each pair of P-EN/E-PG neuron around the ring, if the EB ring is indeed closed, and which might facilitate direct recurrence in close spatial proximity (i.e. a very fast connection).

We further expanded the paragraph to additionally contrast the solution of the two species in closing the ring circuit, i.e. physically closed EB ring in the fruit fly versus neurons cross-innervating the medial glomeruli (across the midline) in the PB in the locust.